# FROM WEAK DATA TO STRONG POLICY: Q-TARGETS ENABLE PROVABLE IN-CONTEXT REINFORCEMENT LEARNING

## ABSTRACT

Transformers trained with offline expert-level data have shown remarkable success in In-Context Reinforcement Learning (ICRL), enabling effective decision-making in unseen environments. However, the performance of these models heavily depends on optimal or expert-level trajectories, making them expensive in various real-world scenarios. In this work, we introduce Q-Target Pretrained Transformers (QTPT), a novel framework that leverages Q-learning instead of supervised learning during the training stage. In particular, QTPT doesn't require optimal-labeled actions or expert trajectories, and provides a practical solution for real-world applications. We theoretically establish the performance guarantee for QTPT and show its superior robustness to data quality compared to traditional supervised learning approaches. Through comprehensive empirical evaluations, QTPT consistently outperforms existing approaches, especially when trained on data sampled with non-expert policies.

## 1 INTRODUCTION

Recent advances in large language models have demonstrated their remarkable ability to perform various tasks in a zero-shot or few-shot manner using in-context learning (Brown et al., 2020; Garg et al., 2023). Building upon this paradigm, transformer-based models have been successfully applied to Reinforcement Learning (RL) settings, showing strong In-Context Reinforcement Learning (ICRL) capabilities (Laskin et al., 2022; Lee et al., 2023; Lin et al., 2024). These models can implicitly capture temporal dependencies within sequences of state-action-reward tuples, enabling them to generalize and make decisions in unseen environments.

Despite impressive results achieved by transformers in offline RL, most existing approaches, such as Algorithm Distillation (Laskin et al., 2022) and Decision Pretrained Transformer (Lee et al., 2023), rely heavily on high-quality datasets. However, these data sets are typically generated by expert policies or contain optimal action labels, which may not be readily available or prohibitively expensive to obtain, limiting the applicability of these ICRL methods in many practical applications.

To address these challenges, we propose Q-Target Pretrained Transformers (QTPT), a novel pre-training framework that replaces supervised behavior cloning with a Q-learning objective. Unlike existing methods that depend on optimal action labels or expert-generated trajectories, QTPT learns to approximate the optimal Q-function directly from offline datasets—even if these datasets are generated by suboptimal policies (see Figure 1). By aligning the pretraining objective with reinforcement learning principles, QTPT reduces the reliance on high-quality data and enhances the robustness and adaptability of transformers in a wide range of decision-making tasks.

Theoretically, we prove that the pretrained transformer can approximate the optimal Q-function and further establish the regret upper bound for QTPT, which can be separated into two distinct components. The first component represents the regret from the inherent sample bias influenced by behavior policies, and the second component stems from the model bias affected by the Rademacher complexity and the size of the training dataset. Such a clear separation builds a novel theoretical foundation to isolate the impacts of sample bias and model bias and sheds light on how QTPT balances leveraging available data and mitigating model bias during pretraining.

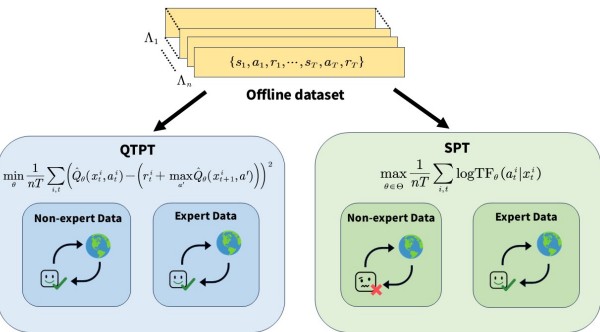

Figure 1: Comparison of Q-target Pretrained Transformers (QTPT) and Supervised Pretrained Transformers (SPT) for in-context RL. While SPT requires expert demonstrations for good performance, QTPT maintains robust performance even when trained on suboptimal data.

We further validate the effectiveness of QTPT through extensive numerical experiments and observe that QTPT consistently outperforms existing supervised-learning-based algorithms, particularly when pretraining datasets are generated by suboptimal policies. This observation highlights QTPT's ability to extract valuable information, even from noisy and suboptimal datasets, where high-quality data is scarce, as in many real-world applications. Achieving strong performance through pretraining on suboptimal data, QTPT lays the foundation for robust and scalable ICRL methods.

The major contributions of this work are summarized as follows:

1. We propose QTPT, a novel framework that pretrains Transformers using Q-learning, reducing the reliance on expert-generated datasets or optimal action labels. By integrating Q-learning with Transformers, we enhance their expressiveness for dynamic programming in Reinforcement Learning (RL), enabling end-to-end value updates. This shift from supervised learning to Q-learning in Transformer pretraining offers a fresh perspective on applying Transformers to RL tasks, marking a significant step in bridging the gap between Transformer models and RL.

2. We theoretically derive the upper bound guarantee on suboptimality for QTPT, demonstrating its superior robustness to data quality over traditional supervised learning approaches. Specifically, we provide strong theoretical guarantees under the finite-horizon Markov Decision Process (MDP) and stochastic linear bandit settings and achieve $\tilde{\mathcal{O}}(T^2/\sqrt{n})$ and $\tilde{\mathcal{O}}(\sqrt{T/n})$ suboptimality gap respectively, where $T$ is the total time steps and $n$ is the sample size (with $\tilde{\mathcal{O}}$ hiding logarithmic factor). QTPT builds on Bellman-error framework by embedding Rademacher-complexity-based generalization into in-context reinforcement learning, extending it from tabular or function-approximation batch RL (Duan et al., 2021) to sequence-model pretraining with end-to-end regret bounds. This approach preserves the crucial decomposition of sample bias and model bias, offering a robust solution for learning from offline data in real-world RL tasks.

3. We conduct extensive experiments that validate our theory and show QTPT outperforms supervised methods, particularly on suboptimal or noisy datasets. This highlights the practical advantages of Q-learning pretraining in environments where high-quality expert data is scarce, further reinforcing the effectiveness of QTPT in real-world applications.

## 1.1 RELATED WORK

**Q-Learning**. Q-learning is a foundational off-policy reinforcement learning algorithm that updates Q-values using the Bellman equation to learn optimal policies through environmental interaction (Watkins & Dayan, 1992; Sutton, 2018). Over the years, a range of advancements have improved Q-learning's efficiency and practical applicability—especially regarding exploration-exploitation trade-offs. Upper-Confidence Bounds (UCB) have enabled more efficient exploration, achieving regret bounds on par with model-based methods (Jin et al., 2018; Zanette & Brunskill, 2019). Optimistic Q-learning algorithms (Even-Dar & Mansour, 2001; Yang et al., 2021) further reduce re-

gret under appropriate conditions, marking a notable improvement over earlier approaches that suffered from poor sample complexity. Deep Q-Networks (DQN) (Mnih et al., 2015) integrates the Q-learning with deep learning to achieve human-level performance in complex tasks. Kapturowski et al. (2018) leverage LSTM networks to better capture temporal dependencies in RL tasks.

Recent advances have adapted Q-learning to large-scale sequence models. Q-SFT (Hong et al., 2024) reframes Q-learning as a supervised fine-tuning problem, optimizing a weighted cross-entropy loss to approximate Q-values and providing theoretical analysis of convergence. Similarly, the Q-learning Decision Transformer (Yamagata et al., 2023) employs a two-step approach: it first relabels reward-to-go (RTG) values using Q-learning, then trains a Decision Transformer (Chen et al., 2021) on these relabeled trajectories.

In contrast, QTPT introduces a novel architectural and algorithmic advance by weaving the entire Bellman backup process directly into the transformer's layers, eliminating the need for external buffers or supervised objectives. Unlike Q-SFT or Q-learning Decision Transformer, QTPT intrinsically performs temporal-difference updates and Bellman backups within its decoder-only architecture, enabling true end-to-end reinforcement learning via self-attention. This unified approach bridges sequence modeling and dynamic programming, allowing QTPT to efficiently handle offline or suboptimal data. Moreover, QTPT is backed by polynomial-in-T regret guarantees, thus making steps in bridging the gap between practice and theory in transformer-based RL.

**Offline Reinforcement Learning**. Offline Reinforcement Learning (RL) focuses on learning optimal policies from pre-collected datasets, without additional interactions with the environment (Levine et al., 2020; Matsushima et al., 2020). One of the major challenges in offline RL is the distribution drift between the behavior policy, which generates the dataset, and the learned policy (Levine et al., 2020; Kostrikov et al., 2021; Rashidinejad et al., 2021). Techniques such as conservative value function estimation, policy constraints, and regularization have been developed to mitigate this challenge (Wu et al., 2019; Kumar et al., 2020; Kidambi et al., 2021; Jin et al., 2022; Dong et al., 2023; Hu et al., 2024; Qu et al., 2024; Setlur et al., 2024; Park et al., 2024). QTPT follows the success of using Transformer models like AD(Laskin et al., 2022) and DPT(Lee et al., 2023), avoiding these issues. The advantages of offline reinforcement learning over behavior cloning (imitation learning) are studied in Kumar et al. (2022). Unlike this stream of work, we utilize *meta-learning framework* to generalize and solve *unseen* RL tasks after pretraining, while offline RL generally focuses on solving *same* RL tasks from which the offline datasets were originally collected.

**In-Context Learning and Reinforcement Learning**. As a non-adaptation method, the In-Context Learning (ICL) approach has shown impressive success in adapting to new tasks by leveraging contextual information from limited examples (Brown et al., 2020; Min et al., 2022; Garg et al., 2023). Various explanations on the mechanism behind ICL has been proposed (e.g., Akyürek et al. (2022); Bai et al. (2023); Von Oswald et al. (2023); Zhang et al. (2023)),gradient descent, temporal difference, and policy updates through their self-attention structures. In-Context Reinforcement Learning (ICRL) addresses sequential decision-making problems by inferring optimal policies from past trajectories. In the realm of ICRL, Wang et al. (2024a) explores how transformers can implement Temporal Difference (TD) learning directly in the forward pass. See Moeini et al. (2025) for a more comprehensive survey. While existing literature mainly focus on the ICRL with supervised pertaining paradigm (e.g., Lin et al. (2024)), our approach leverages Q-learning as the pretraining strategy, explicitly minimizing in-context temporal difference errors. This approach directly aligns the pretraining objective with subsequent in-context decision-making tasks, especially when noisy or suboptimal data are involved.

## 2 MODEL SETTING

### 2.1 PRELIMINARIES

We consider a set of decision-making environments $\mathcal{M}$, each operating over $T$ rounds with shared state and action spaces $(\mathcal{S}, \mathcal{A})$. Each environment $M \in \mathcal{M}$ has a unique transition model $\mathbb{P}_M : \mathcal{S} \times \mathcal{A} \to \Delta(\mathcal{S})$, initial state distribution $\eta_M \in \Delta(\mathcal{S})$, and a reward function $r_M : \mathcal{S} \times \mathcal{A} \to \Delta(\mathbb{R})$. The agent's uncertainty is captured by environment priors $\Lambda_{\text{train}} \in \Delta(\mathcal{M})$ and $\Lambda_{\text{test}} \in \Delta(\mathcal{M})$ for training and testing respectively. This framework represents various scenarios, including $T$ rounds of multi-armed bandit problems and $K$ episodes of $H$-step MDPs with $T = KH$. The training stage

---

**Algorithm 1** Q-target Pretrained Transformers (QTPT)

---

1: **Input**: an Offline Dataset $D$ collected in $\Lambda_{\text{train}}$, and the horizon $T$.
2: // Pretraining model on dataset
3: Randomly initialize $\hat{Q}_\theta$
4: **while** not converged **do**
5:     Sample a batch of $\mathcal{B}$ trajectories from $D$
6:     Compute the Temporal Difference (TD) target $y_t^i = r_t^i + \max_{a' \in \mathcal{A}} \hat{Q}_\theta(s_{t+1}^i, a', D_t^i)$
7:     Compute the loss: $\mathcal{L}_\mathcal{B}(\hat{Q}_\theta) = \frac{1}{|\mathcal{B}|T} \sum_{i=1}^{|\mathcal{B}|} \sum_{t=1}^{T} \left( \hat{Q}_\theta(s_t^i, a_t^i, D_{t-1}^i) - y_t^i \right)^2$
8:     Backpropagate to update $\theta$
9: **end while**
10: // Online test-time deployment
11: Initialize an empty dataset $D$ and sample environment $M \sim \Lambda_{\text{test}}$
12: **for** $t$ in horizon $T$ **do**
13:     Choose $a_t \in \arg\max_a \hat{Q}_\theta(s_t, a, D)$
14:     Execute $a_t$ and observe $r_t, s_{t+1}$
15:     Update $D$ with $(s_t, a_t, r_t)$
16: **end for**

---

uses an offline dataset $D$ consisting $n$ offline trajectories $\{D_T^i = (s_1^i, a_1^i, r_1^i, \ldots, s_T^i, a_T^i, r_T^i)\}_{i=1}^n$, which have been collected by the behavior policy $\pi_\beta$.

We denote a partial interaction trajectory—comprising the sequence of observed states, actions, and rewards—by $D_t = \{s_1, a_1, r_1, \cdots, s_t, a_t, r_t\} \in \mathcal{T}_t = (\mathcal{S} \times \mathcal{A} \times \mathbb{R})^t$. For convenience, we define the *context state* as $x_t = D_{t-1} \cup \{s_t\}$, which lies in the space $\mathcal{X}_t = \mathcal{T}_{t-1} \times \mathcal{S}$. This representation captures the cumulative interaction history up to round $t-1$ and the current state $s_t$. Analogous to standard state transitions in reinforcement learning, the context state evolves based on the current context and action. Specifically, the next context state is given by:

$$x_{t+1} = x_t \cup \{a_t, r_t, s_{t+1}\} \text{ where } r_t \sim r_M(\cdot|s_t, a_t), s_{t+1} \sim \mathbb{P}_M(\cdot|s_t, a_t).$$

We formalize this dynamic with the *context transition model*, defined as $\mathbb{T}_{M,t} : \mathcal{X}_t \times \mathcal{A} \to \Delta(\mathcal{X}_{t+1})$, which maps a given context-action pair to a distribution over the next context state. A *policy* $\pi$ maps a context state $x_t \in \mathcal{X}_t$ to a distribution over the actions $\pi(\cdot|x_t) \in \Delta(\mathcal{A})$.

To simplify the notation, in this work we only consider the no decay-factor setting which is also used in literature (e.g.,Lin et al. (2024)). We define the value function $V_{M,t}^\pi : \mathcal{X}_t \to \mathbb{R}$, the Q-function $Q_{M,t}^\pi : \mathcal{X}_t \times \mathcal{A} \to \mathbb{R}$ for policy $\pi$, and the Bellman operator $\Gamma : \mathbb{R}^{\mathcal{X} \times \mathcal{A}} \to \mathbb{R}^{\mathcal{X} \times \mathcal{A}}$ as follows:

$$V_{M,t}^\pi(x_t) = \mathbb{E}\left[ \sum_{h=t}^{T} r_h \,\middle|\, x_t \right] \tag{1}$$

$$Q_{M,t}^\pi(x_t, a_t) = \mathbb{E}\left[ r_M(s_t, a_t) \right] + \mathbb{E}_{x_{t+1} \sim \mathbb{T}_{M,t}(\cdot|x_t, a_t)} \left[ V_{M,t+1}^\pi(x_{t+1}) \right] \tag{2}$$

$$(\Gamma Q_{M,t})(x_t, a_t) = r_M(s_t, a_t) + \mathbb{E}_{x_{t+1} \sim \mathbb{T}_{M,t}(\cdot|x_t, a_t)} \left[ \max_{a'} Q_{M,t+1}(x_{t+1}, a') \right] \tag{3}$$

Finally, the optimal value function and Q-function are given by $V_{M,t}^*(x_t) = \max_\pi V_{M,t}^\pi(x_t)$ and $Q_{M,t}^*(x_t, a_t) = \max_\pi Q_{M,t}^\pi(x_t, a_t)$ for $\forall x_t \in \mathcal{X}_t, a_t \in \mathcal{A}$ respectively.

## 2.2 Algorithm

This section formally presents the Q-target Pretrained Transformer (QTPT) algorithm. We first construct the approximator for the Q-value function. To learn from the offline dataset $D = \{(s_1^i, a_1^i, r_1^i, \ldots, s_T^i, a_T^i, r_T^i)\}_{i=1}^n$, we utilize a class of sequence functions $\hat{\mathcal{Q}} = (\hat{\mathcal{Q}}_1 \times \cdots \times \hat{\mathcal{Q}}_T)$, where each $\hat{Q}_t \in \hat{\mathcal{Q}}_t$ tries to approximate the optimal Q-value function at time step $t$. For instance, a Transformer can be viewed as a model of sequential $Q$ functions. For simplicity, we denote the full Q-function approximator as $\hat{Q} = (\hat{Q}_1, \ldots, \hat{Q}_T) \in \hat{\mathcal{Q}}$ and optimize it as follows:

$$\min_{\theta \in \Theta} \quad \frac{1}{nT} \sum_{i=1}^{n} \sum_{t=1}^{T} \left( \hat{Q}_t(x_t^i, a_t^i; \theta) - \left( r_t^i + \max_{a'} \hat{Q}_{t+1}(x_{t+1}^i, a'; \theta) \right) \right)^2, \tag{4}$$

where $n$ is the number of trajectories from $D$ and we parameterize the approximated Q-value function is by $\theta$. The formal algorithm of *Q-Target Pretrained Transformer* (QTPT) is stated in Algorithm 1. During the pretraining stage, the approximated Q-value functions are learned in a way that minimizes the differences between the predicted Q-values and the bootstrapped targets. At test time, actions are selected greedily.

## 2.3 ANALYSIS FRAMEWORK

Each approximated function $\hat{Q}_t$ induces a time-dependent greedy policy $\pi_{\hat{Q}_t}$. The overall policy $\pi_{\hat{Q}}$ is defined as the sequence of time-dependent policies $\{\pi_{\hat{Q}_t}\}_{t=1}^T$, greedily selecting actions at each time step. Let $\tilde{Q}_n = \arg\min_{Q \in \hat{Q}} \mathcal{L}_n(Q)$ be the final learned function minimizing expected Bellman error on offline data, where $\mathcal{L}_n(Q) = \mathbb{E}_{D_T^n \sim \pi_\beta}[(Q(x,a) - (\Gamma Q)(x,a))^2]$. Its induced policy $\pi_{\tilde{Q}_n}$ greedily selects actions per $\{\pi_{\tilde{Q}_{n,t}}\}_{t=1}^T$ (omitting $\theta$ for readability). We define the expected cumulative reward for policy $\pi$ is $\mathbb{E}_{M \sim \Lambda, x_1 \sim \eta_M}[V_{M,1}^\pi(x_1)]$, and the optimal expected cumulative reward is $\mathbb{E}_{M \sim \Lambda, x_1 \sim \eta_M}[V_{M,1}^*(x_1)]$. The suboptimality gap quantifies the expected cumulative reward difference between the optimal policy $\pi^*$ and the learned policy $\pi_{\tilde{Q}_n}$. This gap can be decomposed into two components: sample bias and model bias. Specifically:

$$\mathsf{Subopt}_\Lambda(\pi^*, \pi_{\tilde{Q}_n}) = \underbrace{\mathsf{Subopt}_\Lambda(\pi^*, \pi_{\hat{Q}_n^*})}_{\text{Sample Bias}} + \underbrace{\mathsf{Subopt}_\Lambda(\pi_{\hat{Q}_n^*}, \pi_{\tilde{Q}_n})}_{\text{Model Bias}}, \tag{5}$$

where $\hat{Q}_n^* = \arg\min_Q \mathcal{L}_n(Q)$. Sample bias arises from sub-optimal policy decisions due to the static dataset's inability to fully represent the true environment dynamics. Model bias stems from the inherent limitations of both the model architecture and learning algorithm that prevent the learned policy from reaching its theoretical optimal performance.

## 3 MAIN ANALYSIS AND RESULTS

To streamline the analysis of Eq. 5, we denote $d_{M,t}^{\pi^*}(x_t, a_t)$ as the marginal distribution at time $t$ for the optimal policy in the environment $M$, and $d_{M,t}^{\pi_\beta}(x_t, a_t)$ for any behavior policy $\pi_\beta$. Define the marginal occupancy as $d_{M,t}^\pi(x) = \sum_a d_{M,t}^\pi(x, a)$. We first state three standard assumptions in the reinforcement learning literature:

**Assumption 3.1** (Bounded Rewards and Distributional Coverage). *We assume the reward function* $r_M$ *and the marginal state-action distributions* $d_{M,t}^{\pi^*}$ *and* $d_{M,t}^{\pi_\beta}$ *satisfy the following conditions:*

*(a)* ***(Bounded Rewards)*** *The reward function is uniformly bounded:* $|r_M(s,a)| \le 1$ *for all* $(s,a) \in \mathcal{S} \times \mathcal{A}$.

*(b)* ***(Optimal Policy Concentrability)*** *The behavior policy* $\pi_\beta$ *provides sufficient support for the optimal policy* $\pi^*$: *if* $d_{M,t}^{\pi^*}(x_t, a_t) > 0$, *then* $d_{M,t}^{\pi_\beta}(x_t, a_t) > 0$, *for all* $t \in [T]$.

*(c)* ***(Lower-Bounded Occupancy)*** *The occupancy density under* $\pi_\beta$ *is bounded on its support:*

$$L^{-1} := \inf_{(x_t, a_t): d_{M,t}^{\pi_\beta}(x_t, a_t) > 0} d_{M,t}^{\pi_\beta}(x_t, a_t), \quad \forall t \in [T].$$

The bounded rewards assumption ensures that the reward is upper-bounded to avoid trivial solutions. Optimal Policy Concentrability assumes that $d^{\pi_\beta}$ covers the trajectory of some optimal policy $\pi^*$, and Lower-Bounded Occupancy ensures that the positive occupancy density under $\pi_\beta$ is bounded away from 0. Note that these two assumptions (see Assumptions 4.1 and 4.2 in Nguyen-Tang et al. (2023)) ensure that an optimal policy is statistically learnable from offline data and are significantly weaker than the uniform coverage parameter considered in the batch reinforcement learning literature (see Assumption 1 in Duan et al. (2021) and Chen & Jiang (2019)).

We define a Transformer-based Q function approximator $TF_\theta(\cdot)$. Elements in the history $x_t$ are mapped into a fixed-dimensional embedding space, and the candidate action $a$ is appended to the end of this sequence. This token sequence is then fed into a standard decoder-only Transformer architecture, and after processing, the final token produces a contextualized embedding, which is passed through a linear projection layer to produce the scalar prediction $TF_\theta(x_t, a)$.

## 3.1 BOUND OF SAMPLE BIAS

Denote the function bound of $Q \in \hat{\mathcal{Q}}$ as $B_Q = \sup_{x,a} \|Q(x,a) - (\Gamma Q)(x,a)\|_2$. We will use the general notation $\mathcal{L}_n$ to denote the empirical loss calculated on $n$ samples, which corresponds to the $\mathcal{L}_\mathcal{B}$ in our Algorithm on a mini-batch $\mathcal{B}$. We can now state the sample bias bound for QTPT.

**Proposition 3.2.** *(**Bound of Sample bias in** **QTPT**) If for all tuples $(D_{t-1}^i, s_t^i, a_t^i, y_t^i)$, there exists a constant $B > 0$ such that $B_{\hat{Q}_n^*} \leq B$. Per Assumption 3.1, with the probability at least $1 - \delta$, we have*

$$\text{Subopt}_\Lambda(\pi^*, \pi_{\hat{Q}_n^*}) \leq \mathcal{O}\Big(\sqrt{L}T^2\Big[\big(\frac{2\log(2T/\delta)}{n}\big)^{1/4} + \frac{c(\hat{\mathcal{Q}}, n)}{\sqrt{n}}\Big] + \sqrt{L}TB\Big),$$

*where $c(\hat{\mathcal{Q}}, n) = \sqrt{\max\{\sqrt{n \log |\hat{\mathcal{Q}}|}, \log |\hat{\mathcal{Q}}|\}}$.*

Compared to the worst-case generalization bounds in reinforcement learning literature that rely on uniform covering numbers (see (Murphy, 2005)), our error bound improves the dependence on the time horizon $T$ from exponential to polynomial. Specifically, $\sqrt{L}TB$ reflects the approximation bias, matching the error in Assumption 2 of Duan et al. (2021) and extending to in-context, sequence-model bound. We can further specify that $B$ can be chosen to scale with the order of $\mathcal{O}(1/T)$ when the function class possesses suitable complexity and the sample size $n$ is large enough.

## 3.2 BOUND OF MODEL BIAS

We start with the simple stochastic linear bandit problem. At each time step $t = 1, 2, ..., T$, the agent selects an action $a_t \in \mathbb{R}^d$ from a set of actions $\mathcal{A}_t$. Upon taking action $a_t$, the agent receives a reward $r_t = \langle a_t, w^* \rangle + \epsilon_t$, where $w^*$ is an unknown parameter vector, and $\epsilon_t$ represents i.i.d. noise with bounded variance. Without loss of generality, $\|a_t\|_2 \leq 1$. We define the Gram matrix of actions as $G_t = \sum_{i=1}^t a_i a_i^\top$, and show the upper bound for the $TF_\theta$ solved via Eq. 4 in Proposition 3.3.

**Proposition 3.3.** *(**Bound of Model Bias in** **QTPT** ***on Stochastic Linear Bandit***) If the minimum eigenvalue of the Gram matrix $G_t$ satisfies $\lambda_{min}(G_t) \geq \alpha nt$ for some constant $\alpha > 0$, then for the Transfomer $TF_\theta(\cdot)$ at appropriate scale, with probability at least $1 - \delta$, it satisfies*

$$\text{Subopt}(\pi_{\hat{Q}_n^*}, \pi_{\tilde{Q}_n}) \leq \mathcal{O}\left(\sigma\sqrt{T}\sqrt{\frac{d \log T}{\alpha n}}\right).$$

Proposition 3.3 highlights that QTPT's model bias in the stochastic linear bandit setting, illustrating clear dependencies on dimensionality $d$, horizon $T$, and dataset size $n$. This indicates that QTPT effectively leverages offline data, with bias diminishing as data increases.

For finite-horizon MDPs, we establish the upper bound on the model bias term in Proposition 3.4.

**Proposition 3.4.** *(**Bound of Model Bias in** **QTPT** ***on Finite-Horizon MDP***) With probability at least $1 - \delta$, we have*

$$\text{Subopt}_\Lambda(\pi_{\hat{Q}_n^*}, \pi_{\tilde{Q}_n}) \leq \mathcal{O}\Big(T^2\Big(\frac{1}{\sqrt{n}} + \frac{\sqrt[4]{\log T}}{\sqrt[4]{n}}\Big)\Big).$$

Proposition 3.4 demonstrates that the model bias decreases in the sample size $n$ at two distinct rates: $O(n^{-1/2})$ from the statistical complexity and $O((\log T)^{1/4}(n)^{-1/4})$ from the Hoeffding's inequality. This observation suggests that, given a sufficiently large sample size, the model's deviation can be contained with a high probability.

## 3.3 FINAL BOUND

Combining the sample bias bound (Proposition 3.2) and model bias bounds (Propositions 3.3, 3.4), we obtain the following final bound of QTPT.

**Proposition 3.5.** *(**Upper Bound of** **QTPT**) Assume that Assumptions 3.1 holds, for any environment $(\mathcal{S}, \mathcal{A}, T, \mathbb{P}, r, \eta)$, we assume $\Lambda_{test}/\Lambda_{pre} \leq \mathcal{C}$ where $\mathcal{C} > 0$ is a constant. Combining Proposition*

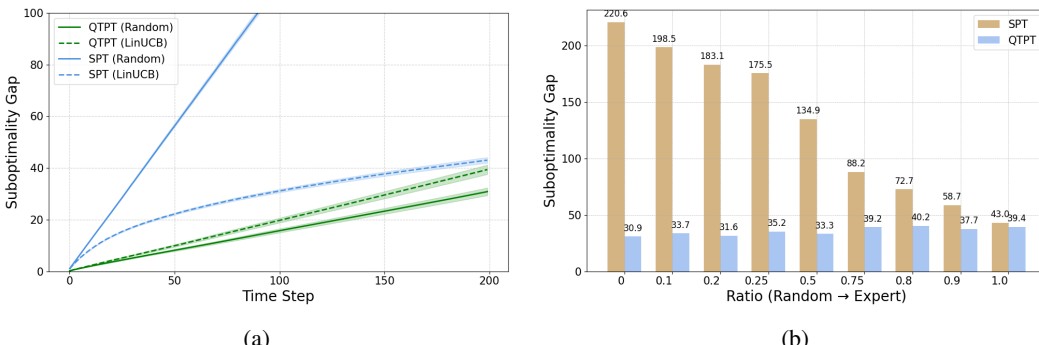

(a)                                                                  (b)

Figure 2: Performance Comparison between QTPT and SPT in Linear Bandit Settings. **Left**: Suboptimality gap over time for QTPT and SPT under Random and LinUCB policies, averaged over 1000 runs. QTPT consistently outperforms SPT, particularly with LinUCB. **Right**: Suboptimality gap across varying data mix ratios (Random → Expert) for offline pretraining. QTPT maintains robust performance with limited expert data, while SPT degrades significantly as expert ratio decreases.

*3.2, 3.3, and 3.4, with probability at least $1 - \delta$, the suboptimality can be bounded as follows:*
*Stochastic Linear bandit:*

$$\mathsf{Subopt}_{\Lambda_{test}}(\pi^*, \pi_{\tilde{Q}_n}) \leq \mathcal{C} \cdot \mathcal{O}\Big( \tfrac{\sqrt{T \log T}}{\sqrt{n}} + \sqrt{L}T^2 \Big[ \tfrac{\sqrt[4]{\log T}}{\sqrt[4]{n}} + \tfrac{c(\hat{\mathcal{Q}}, n)}{\sqrt{n}} \Big] \Big).$$

*Finite-Horizon MDP:*

$$\mathsf{Subopt}_{\Lambda_{test}}(\pi_{\tilde{Q}_n}) \leq \mathcal{C} \cdot \mathcal{O}\Big( \sqrt{L}T^2 \Big[ \tfrac{\sqrt[4]{\log T}}{\sqrt[4]{n}} + \tfrac{c(\hat{\mathcal{Q}}, n)}{\sqrt{n}} \Big] + \tfrac{T^2}{\sqrt{n}} + \tfrac{T^2 \sqrt[4]{\log T}}{\sqrt[4]{n}} \Big).$$

The constant $\mathcal{C}$ defined in Proposition 3.5 describes the environment-level distribution shift between the set of environments used during pretraining $\Lambda_{\text{pre}}$ and those encountered during testing $\Lambda_{\text{test}}$, which can differ in aspects like the state transitions, reward structures and the size of action space, for example. It reflects how well the pretraining data prepares the model for test environments.

Proposition 3.5 establishes theoretical guarantees for QTPT in both stochastic linear bandit and general MDP settings. To better understand these bounds, we compare them with recent work by Lin et al. (2024) Lin et al. (2024), who analyze SPT for stochastic linear bandits. Their analysis decomposes the total regret into three components: statistical estimation bias, approximation bias $\epsilon_{\text{real}}$ (which is similar to $\epsilon_{\text{tf}}$ in our setting, see Section B.4 for details), and intrinsic expert bias $\epsilon_{\text{approx}}$.

**When offline dataset is collected by random policy.** In the stochastic linear bandit framework, when using a purely random action-selection strategy, the regret bound in Theorem 9 of Lin et al. (2024) degrades from $\mathcal{O}(\sqrt{T} \log T)$ to $\mathcal{O}(T)$. Our analysis for the stochastic linear bandit case achieves a tighter bound of $\mathcal{O}(\sqrt{T \log T / n})$, demonstrating that QTPT achieves lower asymptotic regret than supervised-pretraining Transformers when trained on randomly generated data.

**When offline dataset is collected by expert policy.** Even when the offline trajectories are generated by LinUCB with $\mathcal{O}(\sqrt{T} \log T)$ regret, our bound is competitive. The $\sqrt{n}$ factor in our bound corresponds to the generalization term in Theorem 3.2 of Antos et al. (2007), which analyzes batch fitted Q-iteration in continuous-action MDPs. This alignment suggests that our suboptimality bound follows the same inverse relationship with the offline sample size as in prior work.

### 3.4 Data Criterion in which QTPT will outperform SPT

In this section, we analyze the properties of offline datasets and identify the conditions under which QTPT outperforms SPT, focusing on how dataset characteristics influence policy learning. Inspired by (Kumar et al., 2022), we first define the non-informative context set as follows:

**Definition 3.6.** (**Non-informative Context**) For an environment $M$ and its optimal Q-function $Q_M^*$, the non-informative context set $G$ contains all $a \in \mathcal{A}$ and $x \in \mathcal{X}$ such that $\nu_M(x, a) \leq O(T^{-1})$, where $\nu_M(x, a) = \max_{a'} Q_M^*(x, a') - Q_M^*(x, a)$.

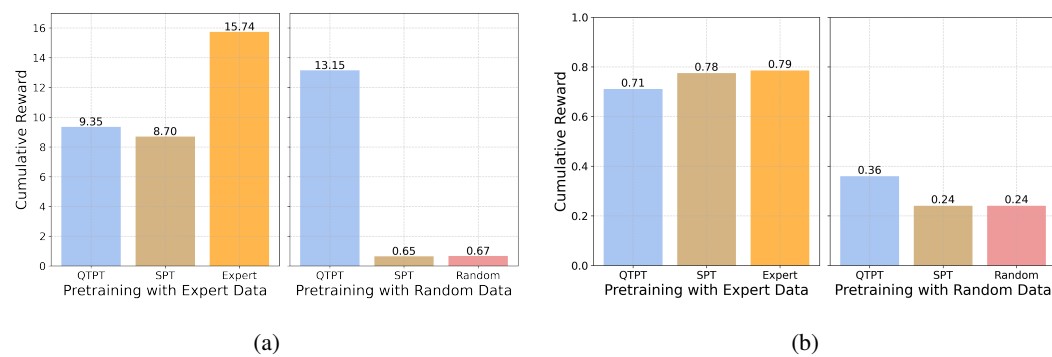

(a)                                                           (b)

Figure 3: **Left**: Comparison of average cumulative rewards (larger cumulative rewards denotes superior performance) between QTPT and SPT, including the performance of expert and random policies in **Darkroom** environment. **Right**: Comparison of average cumulative rewards between QTPT and SPT, including the performance of expert and random policies in **Dark key-to-door**.

For any $(x, a)$ pair in $G$, there is no alternative action $a' \in \mathcal{A}$ such that the value of optimal $Q$ function being significantly improved. In this situation, simply following the behavior policy won't lead to meaningful damage in the output policy, and SPT yields similar behavior to QTPT. However, when there exists a potential action that $\nu_M$ is large enough, via the step 6 in Algorithm 1, QTPT can capture and learn it into the output policy while SPT suffers from memorizing the behavior policy. We formally state this observation in the following Proposition 3.7.

**Proposition 3.7.** *Let $G$ be the non-informative context set of an environment $M$ with optimal $Q$ function $Q_M^*$. Assuming there exists a positive constant $c$ such that for $(x, a) \in (\mathcal{X} \times \mathcal{A}) \setminus G$, $\nu_M(x, a) > c$. Then, under some mild conditions, we have*

$$\mathsf{Subopt}_\Lambda(\pi^*, \hat{\pi}_{QTPT}) \lesssim \mathsf{Subopt}_\Lambda(\pi^*, \hat{\pi}_{SPT}).$$

## 4 NUMERICAL EXPERIMENTS

In this section, we benchmark the performance of the proposed QTPT to existing supervised-learning-based algorithms and validate our theoretical findings through numerical experiments on a stochastic linear bandit setting, markov decision processes, and math reasoning.

### 4.1 STOCHASTIC LINEAR BANDIT

We consider a stochastic linear bandit problem with dimension $d = 5$, arms $A = 10$, and horizon $T = 200$. At each time $t \in [200]$, an agent chooses action $a_t$ and receives reward $r_t = \langle a_t, \theta^* \rangle + \epsilon_t$, where $\epsilon_t \sim N(0, 1.5^2)$ and the parameter $\theta^*$ is sampled from uniform distribution $[0, 1]^d$. The action set $\mathcal{A}_t = \mathcal{A}$ is fixed over time with actions i.i.d. drawn from the uniform distribution $[0, 1]^d$.

**Pretraining Data Collection** For pretraining, we collect two distinct offline datasets: one generated by selecting random actions and the other by using the LinUCB algorithm (see Section C.2 for details). We compile $100,000$ trajectories for both cases.

**Comparison and Implementation** We evaluate the performance of Q-target Pretrained Transformers (ours) and Supervised Pretrained Transformers (SPT). The SPT implementation follows the setup in Lin et al. (2024). Transformer models are based on the GPT-2 architecture (Garg et al., 2022), with 8 layers, 4 attention heads, and an embedding dimension of $D = 256$, using ReLU activation (see Section C.6 for details.)

Figure 2a shows the resulting cumulative regret performance, demonstrating that QTPT outperforms SPT on both random and LinUCB datasets. Additional ablation studies are provided in Section D.1.

We further compare QTPT and SPT under varying data qualities by mixing random and LinUCB-generated trajectories during pretraining. As shown in Figure 2b, the performance of SPT degrades significantly as the proportion of expert data decreases, indicating high sensitivity to data quality. In contrast, QTPT maintains consistently low suboptimality gaps across all mixture ratios, demonstrating strong robustness under suboptimal supervision.

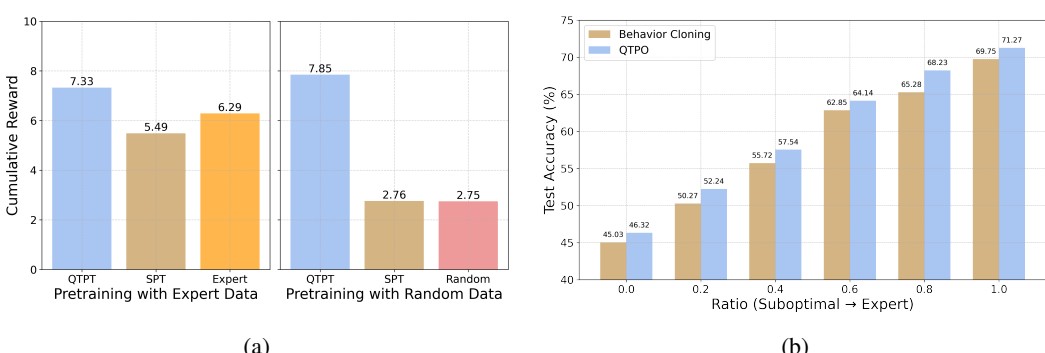

(a)                                                      (b)

Figure 4: **Left**: Comparison of average cumulative rewards between QTPT and SPT, including the performance of expert and random policies in **Miniworld** environment. **Right**: Test accuracy (%) on the GSM8k test set, evaluated across various data mixture ratios (Suboptimal to Expert).

This stems from their training objectives: SPT relies on supervised behavior cloning and is sensitive to demonstration quality, whereas QTPT, trained with Q-learning, leverages diverse trajectories more effectively. Even with suboptimal offline data, Q-learning enables QTPT to generalize better across the state–action space, avoiding sharp performance degradation.

We also conduct experiments to show that when test environment differs from training distribution, QTPT empirically outperforms SPT by a significant margin. See Section D.2 for details.

### 4.2 MARKOV DECISION PROCESSES: DARKROOM, DARK KEY-TO-DOOR, MINIWORLD

We evaluate on three challenging MDPs: Darkroom, Dark Key-to-door, and Miniworld—standard benchmarks for evaluating in-context reinforcement learning models(Laskin et al., 2022; Lee et al., 2023). Results appear in Figures 3a, 3b, and 4a; setup details are in Appendix C.

QTPT outperforms SPT in most settings, except in Dark Key-to-door with expert data. This exception stems from extremely sparse rewards where Q-learning fails to propagate meaningful signals during training (Andrychowicz et al., 2017; Van Hasselt et al., 2018). Unintuitively, random data pretraining often surpasses expert data in Darkroom and Miniworld since random policies explore broader state-action spaces, providing richer Q-learning signals than limited expert trajectories. This supports Weltevrede et al. (2023) on data diversity's role in zero-shot generalization.

### 4.3 MORE COMPLEX TASK: MATH REASONING

To demonstrate the broader applicability of QTPT, we extended our approach to mathematical reasoning tasks. In this domain, we compare QTPO (QTPT with Policy Optimization) against behavior cloning, a standard supervised learning baseline. The specific adaptations for mathematical reasoning and the rationale for using QTPO instead of a direct application of QTPT are detailed in Section D.3. The results, depicted in Figure 4b, show that QTPO consistently outperforms behavior cloning across all evaluated data mixture ratios. QTPO achieves an average accuracy improvement of **1.81%**. This consistent enhancement is particularly noteworthy as our method operates exclusively on a static offline dataset.

## 5 CONCLUSION AND FUTURE WORK

We propose the Q-target Pretrained Transformers (QTPT) algorithm, which learns effectively without expert-level data. We theoretically analyze the optimality of QTPT and show $\tilde{\mathcal{O}}(T^2/\sqrt{n})$ and $\tilde{\mathcal{O}}(\sqrt{T/n})$ in finite-horizon MDP and Linear Bandit problem respectively. We also demonstrate QTPT outperforms SPT approaches when datasets contain specific number of non-informative contexts. Through experiments with both synthetic and real-world datasets, we observe that QTPT consistently outperforms the existing methods, especially when the datasets are noisy or suboptimal. While this work establishes a framework for combining Q-learning with ICRL, future challenges include extending the approach to other RL algorithms (e.g., PPO Schulman et al. (2017) and

GRPO Ramesh et al. (2024)), and investigating the integration of pessimistic regularization techniques. Specifically, exploring how methods like Conservative Q-Learning (CQL) can be combined with our framework is a crucial next step to further mitigate Out-of-Distribution (OOD) Q-value overestimation, thereby enhancing the method's robustness in settings with poor data coverage. addressing the impact of distribution shifts between training and testing, and verifying the performance of QTPT on a broader range of NLP tasks.

**Reproducibility.** The anonymous source code to this project is available in the supplementary. The proofs to the main theory can be found in B.

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

## A   LIMITATIONS

1. There is a lack of further description about the number of distinct functions in the function approximation class $\hat{\mathcal{Q}}$ .

2. The coefficient $\mathcal{C}$ mentioned in Proposition 3.5 represents the difference between the pretraining and test environments. Here, a detailed analysis of the specific expression of this coefficient is lacking.

## B   PROOFS

### B.1   PROOFS RELATED TO THE SAMPLE BIAS

**Definition B.1** (Rademacher complexity). The Rademacher complexity of function class For a generic real-valued function space $\mathcal{F} \subseteq \mathbb{R}^{\mathcal{Y}}$ and $n$ fixed data points $Y = \{y_1, \cdots, y_n\} \in \mathcal{Y}^n$, the empirical Rademacher complexity is defined as follows:

$$\hat{\mathcal{R}}_Y(\mathcal{F}) = \mathbb{E}_\omega\Big[\sup_{f \in \mathcal{F}}\frac{1}{n}\sum_{i=1}^n \omega_i f(y_i)|Y\Big],$$

where $\omega_i$ are independent and identically distributed Rademacher random variables, ranging from $-1$ to $1$ and, thus, representing randomly assigned labels. Let $\rho$ be the distribution of $Y$, we further define the Rademacher complexity $\mathcal{R}_n^\rho(\mathcal{F}) := \mathbb{E}_\rho[\hat{\mathcal{R}}_Y(\mathcal{F})]$, which measures the complexity of $\mathcal{F}$ by the degree to which the functions in class $\mathcal{F}$ are correlated with the random noise $w_i$.

This definition follows a similar structure in Duan et al. (2021). The Rademacher complexity provides a measure of the complexity of our function class, which will be crucial for bounding the sample bias.

#### B.1.1   PROOF OF PROPOSITION 3.2

*Proof.* On the one hand, for a fixed $Q \in \hat{\mathcal{Q}}$ , considering the union bound, by Lemma G.1 and Lemma G.6 in Duan et al. (2021),

$$\Pr\Big[|\mathcal{L}_n(Q) - \mathcal{L}(Q)| \geq \epsilon_n\Big] \leq \delta,$$

where $\epsilon_n = B_Q^2\sqrt{\frac{2\ln(2T/\delta)}{n}} + 2B_Q\mathcal{R}_n^{\pi_\beta}(\hat{\mathcal{Q}}) + B_Q^2$. Thus, with probability at least $1 - \delta$, $|\mathcal{L}_n(Q) - \mathcal{L}(Q)| \leq \epsilon_n$, and

$$\mathcal{L}(\hat{Q}^*) \leq \mathcal{L}_n(\hat{Q}^*) + \epsilon_n \leq \mathcal{L}_n(Q^*) + \epsilon_n \leq \mathcal{L}(Q^*) + 2\epsilon_n,$$

since $\mathcal{L}(Q^*) = 0$, $\mathcal{L}(\hat{Q}^*) \leq 2\epsilon_n$.
On the other hand, by Assumption 3.1, it implies that $d_{M,t}^{\pi_\beta}(x_t, a_t) \geq L^{-1}$, when $d_{M,t}^{\pi_\beta}(x_t, a_t) > 0$. For any $(t, x_t, a_t)$,

$$\frac{d_{M,t}^{\pi^*}(x_t, a_t)}{d_{M,t}^{\pi_\beta}(x_t, a_t)} \leq \frac{1}{d_{M,t}^{\pi_\beta}(x_t, a_t)} \leq L.$$

Thus, using Performance Difference Lemma by Lemma 6.1 in Kakade & Langford (2002), we have

$$\begin{aligned}
\mathsf{Subopt}_\Lambda(\pi^*, \pi_{\hat{Q}^*}) &= \mathbb{E}_{M \sim \Lambda}\textstyle\sum_{t=1}^T \mathbb{E}_{d^{\pi^*}}\Big[|(\Gamma\hat{Q}^*)(x_t, a) - \hat{Q}^*(x_t, a)|\Big] \\
&\leq \sqrt{T\textstyle\sum_{t=1}^T \mathbb{E}_{d^{\pi^*}}\Big[|\hat{Q}^*(x_t, a) - (\Gamma\hat{Q}^*)(x_t, a)|^2\Big]} \\
&\leq T\sqrt{L}\sqrt{\mathcal{L}_n(\hat{Q}^*)} \\
&\leq \sqrt{L}T\sqrt{2\epsilon_n} \\
&\leq \sqrt{L}\Big[2T^2(\frac{2\log(2T/\delta)}{n})^{1/4} + \sqrt{2}T^{3/2}\sqrt{\mathcal{R}_n^{\pi_\beta}(\hat{\mathcal{Q}})} + TB\Big],
\end{aligned}$$

using Proposition 6.1 in literature Duan et al. (2021), we get the final results as

$$\mathsf{Subopt}_\Lambda(\pi^*, \pi_{\hat{Q}^*}) \le \mathcal{O}\Big(\sqrt{L}T^2\Big[(\frac{2\log(2T/\delta)}{n})^{1/4} + \sqrt{\max\{\sqrt{\frac{\log|\mathcal{Q}|}{n}}, \frac{\log|\mathcal{Q}|}{n}\}}\Big] + \sqrt{L}TB\Big).$$

$\square$

## B.2 Proofs Related to the Model Bias

### B.2.1 Proof of Proposition 3.3

*Proof.* At iteration $k$, with current parameters $w^{(k)}$, compute $y_t^{i,(k)} = r_t + \max_{a' \in \mathcal{A}_{t+1}} Q_{w^{(k)}}(D_t^i, s_{t+1}^i, a')$, note that $y_t^{i,(k)}$ is now a fixed constant w.r.t. $w$. Thus, QTPT at round $t-1$ solves the ordinary least-square problem $\hat{w}_{t-1} = \arg\min_w \sum_{i=1}^n \sum_{u=1}^{t-1}(\langle a_u^i, w \rangle - r_u^i)^2$. A greedy policy then selects $a_t = \arg\max_{a \in \mathcal{A}_t} \langle a, \hat{w}_{t-1} \rangle$. Define the estimation error term $EE_t = \sup_{a \in \mathcal{A}_t} |\langle a, w^* - \hat{w} \rangle|$. We get the solution $\hat{w}_{t-1} = G_{t-1}^{-1} \sum_{i=1}^n \sum_{u=1}^{t-1} a_u^i r_u^i$.
On the one hand, by Theorem 2 in Abbasi-Yadkori et al. (2011)Abbasi-yadkori et al. (2011), for zero-mean, $\sigma$-sub-aussian noise $\epsilon_u$, with probability at least $1 - \delta$, we have

$$|\hat{w}_{t-1} - w^*\|_{G_{t-1}} \le \sigma\sqrt{2\log\frac{\det(G_{t-1})^{1/2}}{\det(\alpha n I)^{1/2}\delta}} = \beta_{t-1}$$

On the other hand, for any $a$,

$$|Q_{\hat{w}} - Q^*| = |\langle a, \hat{w} - w^* \rangle| \le \|a\|_{G_{t-1}^{-1}}\|\hat{w} - w^*\|_{G_{t-1}} \le \frac{\beta_{t-1}}{\sqrt{\lambda_{min}(G_{t-1})}} \le \frac{\beta_{t-1}}{\sqrt{\alpha n(t-1)}},$$

since $\det(G_{t-1}) = \mathcal{O}((\alpha n)^d(t-1)^d)$, we have $\beta_{t-1} = \mathcal{O}(\sigma\sqrt{d\ln t})$.
Thus,

$$EE_t = \sup|Q_{\hat{w}} - Q^*| \le \mathcal{O}(\sigma\sqrt{\frac{d\ln t}{\alpha n t}}).$$

The cumulative suboptimality over $T$ steps as

$$\mathbb{E}_{M \sim \Lambda}\Big[\sum_{t=1}^T (\langle a^*, w^* \rangle - \langle a_t, \hat{w} \rangle)\Big],$$

then we have

$$\langle a^*, w^* \rangle - \langle a_t, \hat{w} \rangle = \Big[\langle a^*, w^* \rangle - \langle a^*, \hat{w} \rangle\Big] + \Big[\langle a^*, \hat{w} \rangle - \langle a_t, \hat{w} \rangle\Big] \le MB_t,$$

Finally, we get that

$$\mathsf{Subopt}(\pi_{\hat{Q}^*}, \pi_{\tilde{Q}}) \le 2\mathbb{E}[\sum_{t=1}^T EE_t] + \epsilon_{\mathrm{tf}} \le \mathcal{O}(\sigma\sqrt{T}\sqrt{\frac{d\log(nT)}{\alpha n}}),$$

where $\epsilon_{\mathrm{tf}}$ is the Transformer approximation error (see Section B.4). $\square$

### B.2.2 Proof of Proposition 3.4

*Proof.* For simplicity, we abbreviate $\hat{Q}_{M,t}^\pi$ as $\hat{Q}_t$. Let $\hat{\delta}_t(x, a) = \hat{Q}_t(x, a) - r - \mathbb{E}_{x_{t+1}}\Big[\max_{a'} \hat{Q}_{t+1}(x_{t+1}, a')\Big]$. Similarly, using the definitions of $\tilde{\delta}_t$ and $\delta_t^*$, we can express $\hat{Q}$ to $\tilde{Q}$ and $Q^*$ in terms of $\hat{\delta}_t$. For function $\hat{Q} = (\hat{Q}, \cdots, \hat{Q}_T)$, define

$$\epsilon(\hat{Q}) = \frac{1}{T}\sum_{t=1}^T \mathbb{E}\Big[\hat{Q}_t - r - \mathbb{E}_{x_{t+1}}\max_{a'} \hat{Q}_{t+1}(x_{t+1}, a')\Big]^2 = \frac{1}{T}\sum_{t=1}^T \mathbb{E}\hat{\delta}_t(x, a)^2.$$

Similarly, using the definition of $\epsilon(\tilde{Q})$, we can express $\hat{Q}$ to $\tilde{Q}$ in terms of $\epsilon(\hat{Q})$. Since $\pi_{\tilde{Q}}(x) = \arg\max_{a'} \tilde{Q}(x, a')$, we can show that

$$V_M^{\pi_{\hat{Q}^*}}(x) - V_M^{\pi_{\tilde{Q}}}(x) \leq V^{\pi_{\hat{Q}^*}}(x) - \tilde{Q}_1(x, \pi_{\hat{Q}^*}(x)) + \tilde{Q}_1(x, \pi_{\tilde{Q}}(x)) - V_M^{\pi_{\tilde{Q}}}(x).$$

For any policy $\pi$, we have

$$\tilde{Q}_1(x_1, \pi(x_1)) - V_{M,1}^\pi(x_1)$$

$$= \mathbb{E}\left[\sum_{t=1}^T \left(\tilde{Q}_t(x_t, a_t) - \mathbb{E}_\pi[\tilde{Q}_{t+1}(x_{t+1}, a_{t+1}) + r_t | x_t, a_t]\right) \Big| x_1, \pi\right]$$

$$= \mathbb{E}\left[\sum_{t=1}^T \left(\tilde{Q}_t(x_t, a_t) - r_t - \tilde{Q}_{t+1}(x_{t+1}, a_{t+1})\right) \Big| x_1, \pi\right].$$

We can show that

$$\left|\mathbb{E}\left[\sum_{t=1}^T \tilde{\delta}_t(x_t, a_t) | x_1, \pi\right]\right| \leq \sqrt{T \sum_{t=1}^T \mathbb{E}\left[\tilde{\delta}_t(x_t, a_t)^2 | x_1, \pi\right]} \leq T\sqrt{\epsilon(\tilde{Q})},$$

which implies that

$$\tilde{Q}_1(x_1, \pi_{\tilde{Q}}(x_1)) - V_{M,1}^{\pi_{\tilde{Q}}}(x_1) = \mathbb{E}\left[\sum_{t=1}^T \tilde{\delta}_t(x_t, a_t) | x_1, \pi_{\tilde{Q}}\right],$$

$$\tilde{Q}_1(x_1, \pi_{\hat{Q}^*}(x_1)) - V_{M,1}^{\pi_{\hat{Q}^*}}(x_1) \geq \mathbb{E}\left[\sum_{t=1}^T \tilde{\delta}_t(x_t, a_t) | x_1, \pi_{\hat{Q}^*}\right].$$

Hence,

$$V_{M,1}^{\pi_{\hat{Q}^*}}(x) - V_{M,1}^{\pi_{\tilde{Q}}}(x) \leq \mathbb{E}\left[\sum_{t=1}^T \tilde{\delta}_t(x_t, a_t) | x_1, \pi_{\tilde{Q}}\right] - \mathbb{E}\left[\sum_{t=1}^T \tilde{\delta}_t(x_t, a_t) | x_1, \pi_{\hat{Q}^*}\right] \leq 2T\sqrt{\epsilon(\tilde{Q})}.$$

since $\hat{\delta}_t(x, a)^2 - \tilde{\delta}_t(x, a)^2 \in [-2T^2, 2T^2]$, using Massart Finite Class Lemma, with probability at least $1 - \delta$, there exist a constant $c$ such that for any $\pi$,

$$\mathbb{E}\tilde{\delta}_t(x, a)^2 - \mathbb{E}\hat{\delta}_t(x, a)^2$$

$$\leq \frac{1}{n} \sum_{(x_t, a_t, r_t, x_{t+1}) \sim D_t} (\tilde{\delta}_t(s, a)^2 - \hat{\delta}_t(x, a)^2) + 2R_n((\tilde{\delta}_t)^2 - (\hat{\delta}_t)^2 | \hat{Q}_t) + 2T^2\sqrt{\frac{2\log(2/\delta)}{n}}$$

$$\leq 2T^2\sqrt{\frac{2\log(2/\delta)}{n}} + cTR_n(\hat{Q}_t)$$

$$\leq 2T^2\sqrt{\frac{2\log(2/\delta)}{n}} + 2cT^2\frac{\sqrt{2\log|\mathcal{Q}_t|}}{n}.$$

Since

$$\epsilon(\tilde{Q}) - \epsilon(\hat{Q}) = \frac{1}{T}\sum_{t=1}^T \left(\mathbb{E}\tilde{\delta}_t(x, a)^2 - \mathbb{E}\hat{\delta}_t(x, a)^2\right),$$

since $\epsilon(\hat{Q})$ indicates the capacity of Transformers approximation (see Section B.4 for details), we can get that

$$\epsilon(\tilde{Q}) \leq \epsilon(\hat{Q}) + 2T^2\sqrt{\frac{2\log(2T/\delta)}{n}} + 2cT\sum_{t=1}^T \frac{\sqrt{2\log|\hat{\mathcal{Q}}_t|}}{n}$$

$$\leq \epsilon_{\text{tf}} + 2T^2\sqrt{\frac{2\log(2T/\delta)}{n}} + 2cT\sum_{t=1}^T \frac{\sqrt{2\log|\hat{\mathcal{Q}}_t|}}{n},$$

Thus,

$$V_{M,1}^{\pi_{\hat{Q}^*}}(x_1) - V_{M,1}^{\pi_{\tilde{Q}}}(x_1)$$

$$\leq T^2\left(\sqrt{2}(\frac{2\log(2T/\delta)}{n})^{1/4} + \sqrt{2c\frac{\sqrt{2\sup\log|\hat{\mathcal{Q}}_t|}}{n}}\right) + \epsilon_{tf}$$

$$= \mathcal{O}\left(T^2\left(n^{-1/2} + (\log T)^{1/4}n^{-1/4}\right)\right).$$

$\square$

### B.2.3 PROOF OF PROPOSITION 3.5

*Proof.* Since

$$\int \Lambda_{\text{test}}(M)\mathbb{E}_{x_1 \sim \eta_M}\big[V_{M,1}^*(x_1) - V_{M,1}^{\pi_{\tilde{Q}}}(x_1)\big]dM \le \mathcal{C}\int \Lambda_{\text{pre}}(M)\mathbb{E}_{x_1 \sim \eta_M}\big[V_{M,1}^*(x_1) - V_{M,1}^{\pi_{\tilde{Q}}}(x_1)dM,$$

thus

$$\mathsf{Subopt}_{\Lambda_{\text{test}}}(\pi^*, \pi_{\tilde{Q}}) \le \mathcal{C}\cdot \mathsf{Subopt}_{\Lambda_{\text{pre}}}(\pi^*, \pi_{\tilde{Q}}).$$

When $\Lambda_{\text{pre}} = \Lambda_{\text{test}}$, its upper bound follows by directly adding the results of Proposition 3.2, 3.3 and 3.4. $\qquad\square$

### B.3 PROOFS RELATED TO THE COMPARISON OF QTPT AND SPT

### B.3.1 PROOF OF PROPOSITION 3.7

At first, we need to prove a result on non-informative context states, which is inspired by Lemma B.10 in (Kumar et al., 2022). For simplify, when $G$ be the non-informative context set, define $G_{\mathcal{A}}, G_{\mathcal{X}}$ be the separate sets to represent all unique context states and actions present in $G$, i.e. $G_{\mathcal{A}} = \{a | \exists x, (x,a) \in G\}, G_{\mathcal{X}} = \{x | \exists a, (x,a) \in G\}$.

For an environment $M$ and $\forall a \in \mathcal{A} \setminus G_{\mathcal{A}}$, $\nu_M(x,a) \simeq \Delta_M(x)$. For $(x,a) \in G$, there exists a constant $\epsilon > 0$ that satisfies $\nu_M(x,a) \le \frac{\epsilon}{T}$.

**Lemma B.2.** *Consider a fixed environment $M$, policy $\hat{\pi}_{QTPT}$ is obtained from the QTPT algorithm. There exists a non-informative context set $G$ satisfying $|\mathcal{X} \setminus G_{\mathcal{X}}| \ge n_0$, and let $\forall a \in \mathcal{A}$, $\frac{d_M^{\pi^*}(a|x)}{d_M^{\pi_{\beta}}(a|x)} \le \alpha_0 < 2$, where $d_M^{\pi_{\beta}}(a|x)$ is the conditional action distribution at context state $x$, $\alpha_0$ is a constant. $\tilde{Q}$ is the learned Q-value, since $\tilde{Q}$ can only differ from $Q^*$, there exists $\epsilon_0 > 0$, s.t. $|\tilde{Q}(x,a) - Q^*(x,a)| \le \epsilon_0$. Then, the probability that the policy $\hat{\pi}_{QTPT}$ doesn't choose the action in $G_{\mathcal{A}}$ at context state $x$ is upper bounded as*

$$\mathbb{P}[\hat{\pi}_{QTPT}(x) \notin G_{\mathcal{A}}]$$
$$\le \exp\Big(-n_0\frac{\alpha_0^2}{2(\alpha_0 - 1)} \times [\frac{|G_{\mathcal{A}}|n_\Delta(x)}{n_0} - \frac{1}{\alpha_0}]^2\Big)$$

*where $n(x,a)$ be the expected number of visit $(x,a)$ in dataset $D$, $n_\Delta$ corresponds to the maximum value of $n(x,a)$ s.t.*

$$\frac{\epsilon}{T} \ge \Delta_M(x) - 2\epsilon_0.$$

*Proof.*

$$
\begin{aligned}
\mathbb{P}[\hat{\pi}_{\text{QTPT}}(x) \notin G_{\mathcal{A}}] =\ & \mathbb{P}_M[\exists a \notin G_{\mathcal{A}}, \forall a_g \in G_{\mathcal{A}}, \hat{Q}(x,a) \ge \hat{Q}(x,a_g)] \\
\le\ & \mathbb{P}_M[\exists a \notin G_{\mathcal{A}}, \forall a_g \in G_{\mathcal{A}}, \hat{Q}(x,a) - Q^*(x,a) \ge \hat{Q}(x,a_g) - Q^*(x,a_g) \\
& + \Delta_M(x) - \epsilon/T] \\
\le\ & \mathbb{P}_M[\cap_{a_g \in G_{\mathcal{A}}}\{\epsilon/T \ge \Delta_M(x) - 2\epsilon_0\}] \\
\le\ & \mathbb{P}_M[\cap_{a \in G_{\mathcal{A}}}\{n(x,a) \le n_\Delta\}]
\end{aligned}
$$

where $n_\Delta$ corresponds to the maximum value of $n(x,a)$ s.t.

$$\frac{\epsilon}{T} \ge \Delta_M(x) - 2\epsilon_0$$

consider the action sampled from $d_M^{\pi_\beta}(a|x)$ belong to the set $G_{\mathcal{A}}$ or not,

$$
\begin{aligned}
\mathbb{P}_M[\cap_{a_g \in G_{\mathcal{A}}}\{n(x,a) \le n_\Delta\}] &\le \mathbb{P}_M[\textstyle\sum_{a_g \in G_{\mathcal{A}}} n(x,a) \le |G_{\mathcal{A}}|n_\Delta(x)] \\
&\le \mathbb{P}_M[\frac{\sum_{a_g \in G_{\mathcal{A}}} n(x,a)}{n_0} \le \frac{|G_{\mathcal{A}}|n_\Delta(x)}{n_0}] \\
&\le \exp\Big(-n_0 KL\Big(Bern(\frac{|G_{\mathcal{A}}|n_\Delta(x)}{n_0})||Bern(1/\alpha_0)\Big)\Big)
\end{aligned}
$$

since $KL(p + \epsilon||p) \ge \frac{\epsilon^2}{2p(1-p)}$, if $p \ge 1/2$, $p = 1/\alpha_0$, we have

$$\mathbb{P}_M[\cap_{a \in G_{\mathcal{A}}}\{n(x,a) \le n_\Delta(x)\}] \le \exp\Big(-n_0\frac{\alpha_0^2}{2(\alpha_0 - 1)} \times [\frac{|G_{\mathcal{A}}|n_\Delta(x)}{n_0} - \frac{1}{\alpha_0}]^2\Big)$$

$\qquad\square$

Combining the result of Lemma B.2, for an appropriate value of $c$ and under some mild conditions, we have the result of Proposition 3.7, which states the comparison of QTPT and SPT when a few informative context states are in some given trajectories.

*Proof.* Suppose there exists $p_M \in [0,1]$ s.t. for any policy $\pi$, the average occupancy of states which is not in non-informative set $G$ is bounded as

$$\sum_{t=1}^{T} \sum_{x \notin G_\mathcal{X}} d_{M,t}^\pi(x) \leq p_M.$$

Any given environment $M \sim \Lambda$ such that $\sup_{x \in \mathcal{X}} \{d_M^{\pi^*}(x)/d_M^{\pi_\beta}(x)\} = m$ be a generalization parameter which is independent of $T$, and $L_0 \leq 1 + 1/n$. A non-informative context set $G$ is given such that $|\mathcal{X} \setminus G_\mathcal{X}| \geq n_0$. For learning algorithm that return a policy $\hat{\pi}^*$, the suboptimality is given:

$$
\begin{aligned}
&\mathsf{Subopt}(\pi^*, \hat{\pi}^*) \\
={}& \sum_x d_M^{\pi^*}(x)(V_{M,1}^{\pi^*}(x) - V_{M,1}^{\hat{\pi}^*}(x)) \\
\leq{}& m\Big[ \underbrace{\sum_{x \notin G_\mathcal{X}} d_M^{\hat{\pi}^*}(x)(Q^*(x;\pi^*) - Q^*(x;\hat{\pi}^*))}_{term\ 1} + \underbrace{\sum_{x \in G_\mathcal{X}} d_M^{\hat{\pi}^*}(x)(Q^*(x;\pi^*) - Q^*(x;\hat{\pi}^*))}_{term\ 2} \Big],
\end{aligned}
$$

On the one hand, consider the part of $x \notin G_\mathcal{X}$ (named term 1), we can construct a new MDP where $r(x,a) = r(x, \pi^*(x))$ for all actions and $\forall x \in G_\mathcal{X}$. Denote the suboptimality of this MDP be $\mathsf{Subopt}_{x \notin G_\mathcal{X}}(\pi^*, \hat{\pi}^*)$, then we have

$$\mathsf{Subopt}(\pi^*, \hat{\pi}^*) = \mathsf{Subopt}_{x \notin G_\mathcal{X}}(\pi^*, \hat{\pi}^*)$$

This can be bounded as in Proposition 3.4 , except with the dependence on all context states $\mathcal{X}$ replaced by $\mathcal{X} \setminus G_\mathcal{X}$ . We get that

$$term\ 1 \lesssim p_M T^2 \Big(\frac{1}{\sqrt{n}} + \frac{\sqrt[4]{\log T}}{\sqrt[4]{n}}\Big)$$

On the other hand, consider the part of $x \in G_\mathcal{X}$ (named term 2),

$$term\ 2 = \sum_{x \in G_\mathcal{X}} \sum_{a \in G_\mathcal{A}} W(x,a,\pi^*) + \sum_{x \in G_\mathcal{X}} \sum_{a \notin G_\mathcal{A}} W(x,a,\pi^*)$$

where $W(x,a,\pi^*) = d_M^{\hat{\pi}^*}(x,a)(Q^*(x;\pi^*) - Q^*(x,a))$. Since

$$\sum_{a \in G_\mathcal{A}} W(x,a,\pi^*) \leq d_M^{\hat{\pi}^*}(x)\mathbb{P}[\hat{\pi}^*(x) \in G_\mathcal{A}] \cdot \frac{\epsilon}{T},$$

$$\sum_{a \notin G_\mathcal{A}} W(x,a,\pi^*) \leq d_M^{\hat{\pi}^*}(x)\mathbb{P}[\hat{\pi}^*(x) \notin G_\mathcal{A}] \cdot c,$$

using Lemma B.2, and note that in this case, we are interested in the setting where $L_0 \simeq 1 + \mathcal{O}(1/n)$, we can get that

$$\sum_{x \in G_\mathcal{X}} d_M^{\hat{\pi}^*}(x)\mathbb{P}[\hat{\pi}^*(x) \notin G_\mathcal{A}] \cdot c \lesssim (1 - p_M) f_{\mathrm{QTPT}}(n,c),$$

where $f = c \cdot \exp(-n_0 \times \frac{(n+1)^2}{2n} \times [\frac{|G_\mathcal{A}|n_\Delta(x)}{n_0} - \frac{n+1}{n}]^2)$ is an exponential function of $-n$. Meanwhile, the corresponding term for SPT is

$$\sum_{s \in G_\mathcal{X}} d_M^{\hat{\pi}^*}(s)\mathbb{P}[\hat{\pi}^*(s) \notin G_\mathcal{A}] \cdot c$$

$$\lesssim (1 - p_M) c \cdot \frac{1}{n}$$

$$= (1 - p_M) g_{\mathrm{SPT}}(n,c).$$

By controlling $c$ and $p_M$ for some appropriate values, we have the comparison of QTPT and SPT as follows:

$$\frac{\mathsf{Subopt}_\Lambda(\pi^*, \hat{\pi}_{\mathrm{QTPT}})}{\mathsf{Subopt}_\Lambda(\pi^*, \hat{\pi}_{\mathrm{SPT}})} \simeq \frac{m\Big[p_M T^2(\frac{1}{\sqrt{n}} + \frac{\sqrt[4]{\log T}}{\sqrt[4]{n}}) + (1 - p_M)(f + \epsilon/T)\Big]}{p_M \frac{T}{n} + (1 - p_M)g},$$

since $T$ is finite, combining $f \to \mathcal{O}(e^{-n}), g \to \mathcal{O}(1/n)$, setting $p_M = \frac{1}{T\sqrt{T}}$, we get that

$$\frac{\mathsf{Subopt}_\Lambda(\hat{\pi}_{\mathrm{QTPT}})}{\mathsf{Subopt}_\Lambda(\hat{\pi}_{\mathrm{SPT}})} \simeq \frac{m\left[p_M T^2(\mathcal{O}(n^{1/2}) + \mathcal{O}(n^{3/4})) + (1 - p_M)fn\right]}{p_M T + (1 - p_M)gn}$$

$$\simeq \frac{m\left[\sqrt{T}(\mathcal{O}(n^{1/2}) + \mathcal{O}(n^{3/4})) + (1 - p_M)fn\right]}{1/\sqrt{T} + (1 - p_M)gn}$$

$$\simeq \frac{(1 - p_M)fn}{(1 - p_M)gn}$$

$$\simeq \mathcal{O}(e^{-n}/n)$$

so we get that

$$\mathsf{Subopt}_\Lambda(\hat{\pi}_{\mathrm{QTPT}}) \lesssim \mathsf{Subopt}_\Lambda(\hat{\pi}_{\mathrm{SPT}}).$$

$\square$

### B.4 TRANSFORMERS APPROXIMATION

**Proposition B.3.** *(**Transformers Approximation of** **QTPT***) For any small $\epsilon_{tf} > 0$, there exists a transformer $TF_\theta(\cdot)$ with an appropriate scale, such that we have*

$$\left| TF_\theta(x_t, a) - \tilde{Q}_t(x_t, a) \right| \le \epsilon_{tf}, \quad \forall t \in [T].$$

*Proof.* **Embedding and Extraction Mappings**

For each $t \in [T]$, we construct two tokens

$$h_{2(t-1)} = \begin{bmatrix} s_t \\ 0_{|\mathcal{A}|+1} \\ pos_{2(t-1)} \end{bmatrix} = \begin{bmatrix} h^a_{2(t-1)} \\ h^b_{2(t-1)} \\ h^c_{2(t-1)} \end{bmatrix}, \quad h_{2t-1} = \begin{bmatrix} 0_{|\mathcal{S}|} \\ a_t \\ r_t \\ pos_{2t-1} \end{bmatrix} = \begin{bmatrix} h^a_{2t-1} \\ h^b_{2t-1} \\ h^c_{2t-1} \end{bmatrix}$$

where $pos$ is the position embedding , $s_t, a_t$ are represented using one-hot embedding, $h^b_{2t-1}$ is used to store the policy at time $t$ given current state $s_t$. we add an empty token $h_{2T} = [0_{|\mathcal{S}|+|\mathcal{A}|+1} \quad pos_{2T}]^T$ to store intermediate calculations. We also include in the tokens the position embedding $pos_i = (i, i^2, 1)^T$ for $i \in [2T]$. We define the token matrix $H_t = [h_0, \cdots, h_{2t-1}] \in \mathbb{R}^{D_{dim} \times 2t}$, $D_{dim} = |\mathcal{S}| + |\mathcal{A}| + 4$ for all $t \in [T]$.

**Pretraining**

During pretraining the Transformer $TF_\theta$ takes in $H^{pre}_T = [h_0, \cdots, h_{2T-1}]$ as the input matrix, and generates $H^{post}_T = TF_\theta(H^{pre}_T)$ as the output. from each $t$, suppose the linear extraction map $\mathcal{G}$ , $h^{out}_t = H^{post}_T[:, 2t-1]$, $\hat{Q}_\theta(x_t, a_t) = [\mathcal{G}h^{out}_t]_{a_t}$, where $[\mathcal{G}h]_a$ be the logit of line $a$.
Calculating $y_t = r_t + \max_{a'} \hat{Q}_\theta(x_{t+1}, a')$, we then updating the parameter $\theta \in \Theta$ by gradient descent method.

**Rollout**

At online deployment time, we initialize an empty context dataset $D_0$. At each timestep $t \in [T]$, given the accumulated context $D_{t-1}$ and the current state $s_t$, we construct the token matrix $H^{pre}_{roll,t} = [h_0, \cdots, h_{2(t-1)-1}, h_{2(t-1)}]$, and generates $H^{post}_{roll,t} = TF_{\hat{\theta}}(H^{pre}_{roll,t})$, from the last output token corresponding to the current state, $h^{out}_{roll,t} = H^{post}_{roll,t}[:, 2t-1]$, we compute the action logits via the extraction map $\mathcal{G}$ as $\{[\mathcal{G}_a h^{out}_{roll,t}]\}_{a \in \mathcal{A}}$. The agent selects the action $a_t$ according to a greedy policy over these logits $a_t = \arg\max_a [\mathcal{G}_a h^{out}_{roll,t}]$. After selecting, the agent executes the action, observes the reward $r_t$ and the next state $s_{t+1}$, then construct the new token $h_{2(t-1)+1}$ and append it to the context dataset $D_t$.
Given the input token matrix $H^{pre}_{roll,t}$, we construct a Transformer that implements the following steps on the last token. For each token $h^{pre}_{2t-1}$, the Transformer implements the following step-by-step structured transformations:

$$
\begin{bmatrix} h_{2t-1}^{pre,a} \\ h_{2t-1}^{pre,b} \\ h_{2t-1}^{pre,c} \end{bmatrix} \xrightarrow{\text{Step 1}} \begin{bmatrix} h_{2t-1}^{pre,a} \\ 0_{\mathcal{A}} \\ r_t \\ pos_{2t-1} \end{bmatrix} \xrightarrow{\text{Step 2}} \begin{bmatrix} h_{2t-1}^{pre,a} \\ \max_{a'} \hat{Q}_\theta(x_{t+1},a') \\ r_t \\ pos_{2t-1} \end{bmatrix}
$$

$$
\xrightarrow{\text{Step 3}} \begin{bmatrix} h_{2t-1}^{pre,a} \\ y_t = r_t + \max_{a'} \hat{Q}_\theta(x_{t+1},a') \\ \star \\ pos_{2t-1} \end{bmatrix} \xrightarrow{\text{Step 4}} \begin{bmatrix} h_{2t-1}^{post,a} \\ h_{2t-1}^{post,b} \\ h_{2t-1}^{post,c} \end{bmatrix}
$$

given the current context $x_t = (D_{t-1}, s_t)$, the Transformer $TF_\theta(\cdot)$ outputs a token $h_{2t-1}^{out}$, from which the Q-values $\hat{Q}_\theta(x_t, a)$ are extracted. Each step is defined as below:

**Step 1 (Reward Extraction)**

There exists a attention-only Transformer $\text{TF}_\theta(\cdot)$ that implements Step 1.

**Proof of Step 1** We prove this step by constructing a Transformer that add $r_t$ from $h_{2t-1}^b$ to $h_{2t+1}^b$. We can construct a two-layer attention-only Transformer with $Q_{1,2}^{(1)}, K_{1,2}^{(1)}, V_{1,2}^{(1)}$, such that for all $i \le 2t-1$,

$$
Q_1^{(1)} h_i^{(0)} = \begin{bmatrix} 1 \\ i \end{bmatrix}, \quad K_1^{(1)} h_i^{(0)} = \begin{bmatrix} i+2 \\ -1 \end{bmatrix}, \quad V_1^{(1)} h_{2t-1}^{(0)} = \begin{bmatrix} 0_{|\mathcal{S}|} \\ 0_{|\mathcal{A}|} \\ r_t \end{bmatrix}, \quad V_1^{(1)} h_{2t}^{(0)} = \begin{bmatrix} 0_{|\mathcal{S}|} \\ 0_{|\mathcal{A}|} \\ 0 \end{bmatrix},
$$

and we choose $Q_2^{(1)} = Q_1^{(1)}, V_2^{(1)} = -V_1^{(1)}, K_2^{(1)} h_i^{(0)} = \begin{bmatrix} i+1 \\ -1 \end{bmatrix}$, summing up the heads, we obtain the update on a subset of coordinates in $h_{2t+1}^{(0),b}$ as

$$
0_{|\mathcal{S}|+|\mathcal{A}|+1} \to 0_{|\mathcal{S}|+|\mathcal{A}|+1} + \sum_{j=1}^{2} \sum_{i=1}^{2t+1} \sigma(\langle Q_j^{(1)} h_{2t+1}^{(0)}, K_j^{(1)} h_i^{(0)} \rangle) V_j h_i^{(0)} = \frac{1}{2t+1} \begin{bmatrix} 0_{|\mathcal{S}|} \\ 0_{|\mathcal{A}|} \\ r_t \end{bmatrix}
$$

We then use another attention layer to multiply the updated vectors by a factor of $2t+1$.

Choosing $Q_1^{(2)} h_i^{(1)} = \sqrt{2t+1} \cdot e_i, K_1^{(2)} h_j^{(1)} = \sqrt{2t+1} \cdot e_j, V_1^{(2)} h_{2t+1}^{(1)} = \frac{1}{2t+1} \begin{bmatrix} 0_S \\ 0_A \\ r_t \end{bmatrix}$, and noting that $\langle Q_1^{(2)} h_i^{(1)}, K_1^{(2)} h_j^{(1)} \rangle = 2t+1$, when $j=i$ and otherwise 0.

**Step 2 (Future Q-Value Lookup)**

Attend to the next state token $h_{2t}^{\text{pre}}$, and extract the maximum predicted $\max_{a'} \hat{Q}_\theta(x_{t+1}, a')$. There exists a Transformer $\text{TF}_\theta(\cdot)$ that implements Step 2.

**Proof of Step 2**

Given $(\hat{Q}_\theta)_T = 0$, we start with constructing an-attention layer, and $\{Q_{jt,s}\}_{s=1}^2, \{K_{jt,s}\}_{s=1}^2, \{V_{jt,s}\}_{s=1}^2$ such that for all $i \le 2t-1$ and $j \le i$,

$$
Q_{jt,1}^{(1)} h_i^{(0)} = \begin{bmatrix} x_t \\ -i \\ 3T \end{bmatrix}, \quad K_{jt,1}^{(1)} h_i^{(0)} = \begin{bmatrix} \hat{Q}_t(\cdot, a_j) \\ 3T \\ j \end{bmatrix}, \quad V_{jt,1}^{(1)} h_i^{(0)} = \begin{bmatrix} 0 \\ ie_{jt} \\ 0 \end{bmatrix}, \quad Q_{jt,2}^{(1)} h_i^{(0)} = \begin{bmatrix} -x_t \\ -i \\ 3T \end{bmatrix},
$$

$K_{jt,2}^{(1)} = K_{jt,1}^{(1)}, V_{jt,2}^{(1)} = -V_{jt,1}^{(1)}$.
where $e_{jt}$ is a one-hot vector supported on some entry of $h^c$. Summing up two heads gives the update for $i \le 2t-1$,

$$
0 \to 0 + \Big[ \sigma(\langle Q_{jt,1}^{(1)} h_i^{(0)}, K_{jt,1}^{(1)} h_i^{(0)} \rangle) - \sigma(\langle Q_{jt,2}^{(1)} h_i^{(0)}, K_{jt,2}^{(1)} h_i^{(0)} \rangle) \Big] e_{jt} = \hat{Q}_t(x_t, a_j) e_{jt}
$$

Denote the resulting token vector by $h_i^{(1)}$.

Next, we construct a MLP layer, s.t for any $x \in \mathcal{X}$ on the corresponding coordinates we have

$$
W_1^{(2)} h_i^{(1)} = \begin{bmatrix} \vdots \\ \hat{Q}_t(x, a_1) \\ \hat{Q}_t(x, a_2) - \hat{Q}_t(x, a_1) \\ \vdots \\ \hat{Q}_t(x, a_{|\mathcal{A}|}) - \hat{Q}_t(x, a_{|\mathcal{A}|-1}) \\ \vdots \end{bmatrix}
$$

where $a_k$ denotes the $k^{th}$ action, and

$$
W_2^{(2)} \sigma(W_1^{(2)} h_i^{(1)}) = \sigma(\hat{Q}_t(x, a_1)) + \sum_{k=2}^{A} \sigma(\hat{Q}_t(x, a_k) - \hat{Q}_t(x, a_{k-1})) = \max_{a \in A} \hat{Q}_t(x, a).
$$

**Step 3 (TD Target Construction)**

Combine $r_t$ and $\max_{a'} \hat{Q}(x_{t+1}, a')$ to form the TD target:

$$
y_t = r_t + \max_{a'} \hat{Q}(x_{t+1}, a').
$$

There exists a Transformer $TF_\theta(\cdot)$ that implements Step 3.

**Proof of Step 3**

The proof is similar to that of the literature Wang et al. (2024a). Let

$$
Q_1^{(1)} h_i^{(0)} = K_1^{(1)} h_i^{(0)} = \begin{bmatrix} 0 \\ 1 \\ 0 \end{bmatrix}, \quad V_1^{(1)} h_{2t-1}^{(0)} = \begin{bmatrix} 0_{|\mathcal{S}|} \\ r_t \\ 0 \end{bmatrix},
$$

$$
Q_2^{(1)} h_i^{(0)} = K_2^{(1)} h_i^{(0)} = \begin{bmatrix} 0 \\ 0 \\ 1 \end{bmatrix}, \quad V_2^{(1)} h_{2t}^{(0)} = \begin{bmatrix} 0_{|\mathcal{S}|} \\ \max_{a'} \hat{Q}_\theta(x_{t+1}, a') \\ 0 \end{bmatrix},
$$

then we have that

$$
h_{2t-1}^{(1)} = \sum_{j=1}^{2} \sum_{i=1}^{2t} \sigma(\langle Q_j^{(1)} h_{2t-1}^{(0)}, K_j^{(1)} h_i^{(0)} \rangle) V_j^{(1)} h_i^{(0)} = \begin{bmatrix} 0_{|\mathcal{S}|} \\ r_t \\ \max_{a'} \hat{Q}_\theta(x_{t+1}, a') \\ 0 \end{bmatrix},
$$

define a MLP layer with $W_1^{(2)} = \begin{bmatrix} 0 & 0 & 0 \\ 0 & 1 & 0 \\ 0 & 0 & 1 \end{bmatrix}$, $W_2^{(2)} = [0 \ 1 \ 1]$. We can get that
$W_2^{(2)} \sigma(W_1^{(2)} h_{2t-1}^{(1)}) = r_t + \max_{a'} \hat{Q}_\theta(x_{t+1}, a')$.

**Step 4 (Q-Value Update):**

There exists a small MLP head (two-layer feed-forward network) which updates $h_{2t-1}$ to regress the Q-value prediction $\hat{Q}(x_t, a_t)$ towards the TD target $y_t$.

**Proof of Step 4**

The proof is related to the literature von Oswald et al. (2023). Let

$$
Q_1^{(1)} h_i^{(0)} = K_1^{(1)} h_i^{(0)} = \begin{bmatrix} 0 \\ 1 \\ 0 \\ 0 \end{bmatrix}, \quad V_1^{(1)} h_{2t-1}^{(0)} = \begin{bmatrix} 0_{|\mathcal{S}|} \\ \hat{Q}_\theta(x_t, a_t) \\ 0 \\ 0 \end{bmatrix},
$$

$$Q_2^{(1)} h_i^{(0)} = K_2^{(1)} h_i^{(0)} = \begin{bmatrix} 0 \\ 0 \\ 1 \\ 0 \end{bmatrix}, \quad V_2^{(1)} h_{2t-1}^{(0)} = \begin{bmatrix} 0_{|\mathcal{S}|} \\ 0 \\ y_t \\ 0 \end{bmatrix}$$

combining two attention heads, we get that $h_{2t-1}^{(1)} = \begin{bmatrix} 0_{|\mathcal{S}|} \\ \hat{Q}_\theta(x_t, a_t) \\ y_t \\ 0 \end{bmatrix}$. define a MLP layer with

$$W_1^{(2)} = \begin{bmatrix} \cdots \\ 0 & 1 & 0 & 0 \\ 0 & 0 & 1 & 0 \\ \cdots \end{bmatrix}, W_2^{(2)} = [0 \quad 1-\alpha \quad \alpha \quad 0], \text{ where } \alpha \text{ be a coefficient. We have that}$$

$$W_2^{(2)} \sigma(W_1^{(2)} h_{2t-1}^{(1)}) = (1-\alpha)\hat{Q}_\theta(x_t, a_t) + \alpha y_t.$$

$\square$

## C  EXPERIMENT DETAILS

### C.1  COMPUTING RESOURCES

All experiments are conducted on two NVIDIA A100 GPUs (40GB each).

### C.2  LINUCB ALGORITHM

Let $T$ denote the time horizon and $\lambda, \alpha > 0$ be input parameters. At each time step $t \in \{1, 2, ..., T\}$, the LinUCB algorithm operates through the following steps:

1. Compute the ridge estimator for the weight vector:

$$\mathbf{w}_{\text{ridge},\lambda}^t = \arg\min_{\mathbf{w} \in \mathbb{R}^d} \left( \frac{1}{2t} \sum_{j=1}^{t-1} (r_j - \langle \mathbf{a}_j, \mathbf{w} \rangle)^2 + \frac{\lambda}{2t} \|\mathbf{w}\|_2^2 \right)$$

2. For each action $i \in [A]$, compute the upper confidence bound:

$$v_{t,i}^* = \langle \mathbf{a}_{t,i}, \mathbf{w}_{\text{ridge},\lambda}^t \rangle + \alpha \sqrt{\mathbf{a}_{t,i}^\top \mathbf{A}_t^{-1} \mathbf{a}_{t,i}},$$

where $\mathbf{A}_t = \lambda \mathbf{I}_d + \sum_{j=1}^{t-1} \mathbf{a}_j \mathbf{a}_j^\top$.

3. Select the action $a_{t,j}$ by:
$$j := \arg\max_{i \in [A]} v_{t,i}^*.$$

### C.3  DARKROOM ENVIRONMENT

Darkroom is considered as a complex Markov decision process (MDP) problem and is utilized as a standard benchmark for evaluating in-context learning Laskin et al. (2022); Lee et al. (2023). In this experiment, the agent must locate an unknown goal within a $10 \times 10$ darkroom. The agent receives a reward of 1 only when it reaches the goal. At each step, the agent can choose from five possible actions: move up, down, left, right, or stay still. If the agent is not at the goal, it receives a reward of 0. The horizon for the Darkroom environment is set to 100 steps. We summarize the details as follows:

**Pretraining Data Collection**  Similar to the stochastic linear bandit problem, we consider two types of policies: A random policy, which selects an action randomly at each position, and an expert policy, which chooses legal actions to avoid crashing into walls and will stay still once the agent receives a reward of 1. To test whether the pretrained model can generalize to **unseen** RL problems in context, we collect datasets from 80 out of the total 100 goals, reserving the remaining 20 for testing. For each training goal, we run both the random and expert policies, collecting 1k trajectories from each policy. This leads to a total of 80k trajectories for the random policy and 80k trajectories for the expert policy, resulting in 160k context trajectories in the pretraining dataset.

**Comparison and Implementation** We evaluate QTPT and SPT under two settings: pretraining on purely random data and on purely expert data. The models are evaluated on an **unseen** task, where the target goal is not included in the pretraining dataset. The model architecture is identical to that used in the stochastic linear bandit experiments (see Section C.6).

## C.4 DARK KEY-TO-DOOR ENVIRONMENT

The Dark Key-to-Door environment extends the complexity of the Darkroom setting by introducing a hierarchical task structure with sparse rewards. In this environment, the agent must sequentially accomplish two objectives: first locate an invisible key to receive a reward of $r = 1$, then find and open an invisible door to receive an additional reward of $r = 1$. This creates a challenging sparse reward scenario where the agent must learn to complete subtasks in the correct order. The environment consists of a $9 \times 9$ grid, and each episode is limited to 50 steps. At the beginning of each episode, the agent is reset to position $(0, 0)$, while the key and door locations are randomly generated across different episodes. Since door and key can be placed in any positions, there are total $81 \times 81 = 6561$ distinct tasks.

**Pretraining Data Collection** To capture diverse exploration strategies, we collect our pretraining data using two distinct behavioral policies. The expert policy employs a systematic spiral search pattern, ensuring comprehensive coverage of the environment while efficiently locating the key and door. In contrast, the random policy selects actions uniformly at random, providing diverse but generally suboptimal exploration patterns. To ensure a balanced representation of both behaviors, we collect 80k trajectories from each policy type. The pretraining dataset is partitioned into 5249 tasks, which represents approximately 80% ($65610.8 \approx 5249$) of the total tasks. The remaining 1312 tasks are held out for testing and are thus considered **unseen**.

**Comparison and Implementation** We evaluate the performance of our QTPT and SPT models under two distinct pretraining configurations: one trained exclusively on random policy data and the other trained exclusively on expert policy data. The evaluation, conducted on **unseen tasks**, specifically assesses the models' ability to generalize the underlying hierarchical task structure and adapt to new key-door configurations during in-context learning. To ensure a fair comparison across environments and configurations, the model architecture is kept consistent with that of previous experiments.

## C.5 MINIWORLD ENVIRONMENT

To evaluate QTPT's effectiveness in visual domains, we employ the Miniworld (Lee et al., 2023) navigation benchmark, which presents a visually grounded navigation task. In this environment, agents must navigate to the correct colored target box among four boxes positioned at the corners of the environment. The agent receives $25 \times 25$ RGB observations with directional conditioning, providing rich visual information that requires the model to process high-dimensional sensory input. The agent has access to three discrete actions: turn left, turn right, and move forward, which creates a more realistic navigation scenario compared to grid-world abstractions. The agent receives a reward of $r = 1$ only when positioned near the target box, with all other states yielding zero reward. Each episode is constrained to 50 timesteps, requiring efficient navigation strategies.

**Pretraining Data Collection** We construct our pretraining dataset using two complementary data collection strategies to capture both optimal and suboptimal navigation behaviors. The expert policy demonstrates intelligent navigation by selecting legal actions that avoid wall collisions and environmental boundaries, and crucially, the agent remains stationary once it successfully reaches the target and receives the reward signal. This policy represents efficient goal-directed behavior in the visual navigation domain. The random policy, conversely, selects actions uniformly at random, generating diverse but often inefficient exploration trajectories that cover various parts of the state space. We collect 24k trajectories from each policy type for training.

**Comparison and Implementation** The evaluation protocol examines QTPT and SPT performance under both pure random data pretraining and pure expert data pretraining conditions. The

model architecture incorporates appropriate visual encoding mechanisms while maintaining consistency with the core transformer structure used in other experimental settings.

### C.6 Implementation Detail

Both Transformer models are based on the GPT-2 architecture (Garg et al., 2022), featuring 8 layers, 4 attention heads, and an embedding dimension of $D = 256$, with ReLU activation functions. The training process is implemented with a batch size of 32 using the Adam optimizer. The learning rate schedule follows a cosine decay with a linear warmup for the first 2,000 training steps, where the peak learning rate is set to $5 \times 10^{-6}$ and the minimum learning rate is $1 \times 10^{-7}$. Weight decay is set to 0.001.

Each model is pretrained for 50 epochs on the collected datasets. After pretraining, the models are evaluated on a test set to assess their effectiveness in the context of the stochastic linear bandit problem.

## D  Additional Experiments

### D.1  Ablation Study on the Stochastic Linear Bandit Problem

We also consider using 'softmax' operation when computing the Q-target in experiments. To evaluate the robustness of QTPT, we conducted ablation experiments examining several key factors: The impact of using either 'softmax' or 'hardmax', the effect of incorporating a double DQN framework for network updates, and the differences between ReLU and softmax activation functions in computing attention scores. Note that our original implementation combines 'softmax', ReLU, and a double DQN framework.

Table 1: Comparison of results for different configurations of QTPT.

| Random Data Results | Default | Double DQN | Softmax + Hardmax | ReLU-Attn + Softmax-Attn |
|---|---|---|---|---|
| Results | 30.9 | 34.01 | 32.48 | 197.26 |
| **Expert Data Results** | Default | Double DQN | Softmax + Hardmax | ReLU-Attn + Softmax-Attn |
| Results | 39.4 | 45.76 | 53.73 | 203.74 |

In addition, we vary the embedding dimension, number of layers, and number of attention heads to evaluate the model's performance under different configurations. Note that the original configuration consists of 8 layers, 4 attention heads, and an embedding dimension of 256, pretrained on purely random data.

Table 2: Results for varying embedding dimensions.

| Embedding Dimension | 16 | 32 | 64 | 128 | 256 |
|---|---|---|---|---|---|
| Results | 135.7 | 54.0 | 40.5 | 33.3 | 30.9 |

Table 3: Results for varying number of attention heads.

| Number of Heads | 1 | 2 | 4 | 8 | 16 |
|---|---|---|---|---|---|
| Results | 34.2 | 30.2 | 30.9 | 29.3 | 31.4 |

### D.2  Experiments on Non-stationary Environment.

We aim to demonstrate the robustness of QTPT by evaluating its performance in non-stationary environments.

Table 4: Results for varying number of layers.

| Number of Layers | 1 | 2 | 4 | 8 |
|---|---|---|---|---|
| Results | 122.3 | 33.6 | 31.0 | 30.9 |

**Experiment Setup:** The pretraining phase remains consistent with the process described in Section 4. Yet, in the test phase, we introduce two different settings:

- **Stationary:** The parameter $\theta^*$ is sampled from a uniform distribution $[0,1]^d$, which matches the conditions of the pretraining stage;

- **Non-stationary:** The parameter $\theta^*$ is sampled from a standard Gaussian distribution $N(0,1)^d$ and rescaled to lie within the range $[0,1]^d$. The rescaling is performed using the transformation:

$$\theta_{\text{scaled}} = \frac{\theta - \theta_{\min}}{\theta_{\max} - \theta_{\min}},$$

  where $\theta_{\min}$ and $\theta_{\max}$ represent the minimum and maximum values of the sampled $\theta$. In our experimental setup, we sample 1000 values from the Gaussian distribution and take the minimum and maximum from these samples for rescaling.

Figure 5 demonstrates that QTPT achieves superior performance in the non-stationary environment compared to the stationary setting. While SPT maintains relatively consistent performance across both environments, it significantly underperforms compared to Q-learning, especially under non-stationary conditions. These results highlight QTPT's enhanced adaptability to changing environmental dynamics and robustness to distributional shifts in the underlying parameters. Notably, the performance gap between QTPT and SPT widens considerably in the non-stationary setting, with QTPT maintaining strong performance while SPT's effectiveness diminishes.

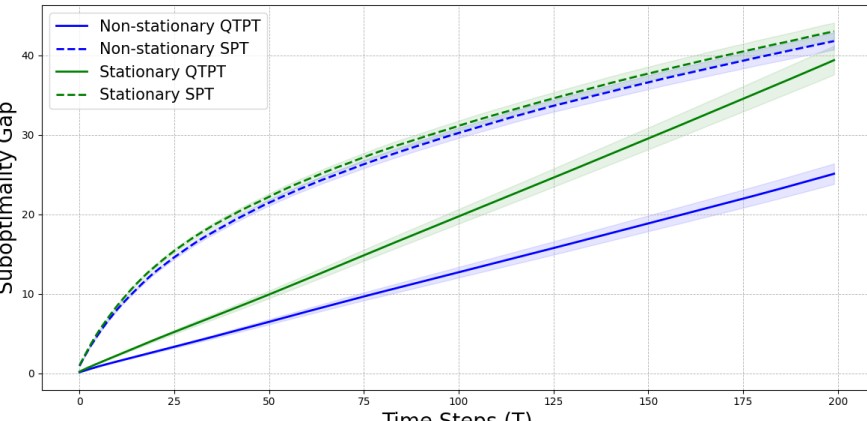

Figure 5: Comparison of Pretrained Transformer Performance Using Q-learning and Supervised Learning in Stationary and Non-stationary Environments. The shaded regions represent the standard deviation of the suboptimality estimates based on 1000 simulation runs.

### D.3 MORE COMPLEX TASK: MATH REASONING

To demonstrate the broader applicability of QTPT, we extended our approach to mathematical reasoning tasks. Each math problem serves as a distinct environment, where generated tokens represent states, the full sequence forms the context-state, and token selection constitutes actions. The reward signal is derived from binary correctness or step-by-step reasoning accuracy. Our method showed consistent performance improvements over supervised learning baselines in these tasks.

A key challenge emerged during preliminary experiments is that when learning a Q-function from offline data alone, and only use the Q-function to generate token, the model frequently produced

---

**Algorithm 2** QTPT with Policy Optimization (QTPO)

---

1: Estimate behavior policy $\hat{\pi}_\beta$ by behavior cloning on offline dataset $D$
2: Training the Value function $\hat{V}_{\pi_\beta}$ and advantage function $\hat{A}_{\pi_\beta}$ together by dueling architecture
   using QTPT on offline dataset $D$.
3: Initialize $k = 0$ and set $\pi_k \leftarrow \hat{\pi}_\beta$
4: **for** $i = 0, 1, 2, \cdots, I$ **do**
5:     Update the policy $\pi$ by maximizing $L_k(\pi)$
6:     **if** $J(\pi) > J(\pi_k)$ **then**
7:        Set $k = k + 1 \& \pi_k \leftarrow \pi$
8:     **end if**
9: **end for**

---

degenerate outputs (e.g., generating repetitive tokens). We hypothesize that this is due to semantic distortion caused by the Bellman update. Specifically, the output in language models consists of token probabilities, and applying cumulative reward-based reasoning to these probabilities disrupts their linguistic coherence, as it is unconventional to treat them as a direct sum of the reward and the probability of the next token.

To address this challenge, we adopt an actor-critic framework in which the Q-function learned by QTPT is not used directly for token generation but to refine the policy. In the offline setting, we modified the Proximal Policy Optimization (PPO) algorithm, drawing inspiration from Behavior Proximal Policy Optimization (BPPO)Zhuang et al. (2023), and incorporated an iterative refinement process, which we name *QTPT with Policy Optimization*(QTPO). The details are provided in Algorithm 2. where

$$L_k(\pi) = \mathbb{E}_{s \sim \rho_{\mathcal{D}}(.), \, a \sim \pi_k(.||s)} \left[ \min \left( \frac{\pi(a|s)}{\pi_k(a|s)} \hat{A}(s,a), \operatorname{clip} \left( \frac{\pi(a|s)}{\pi_k(a|s)}, \, 1 - \epsilon, \, 1 + \epsilon \right) \hat{A}(s,a) \right) \right] \tag{6}$$

and the objective $J(\pi)$ is the total accuracy on the offline dataset.

**Data Preparation** We evaluated our method on the GSM8K dataset Cobbe et al. (2021), utilizing reward labels and training data from Math-Shepherd Wang et al. (2024b). The experimental setup involved two distinct data categories:

- **Expert Data**, consisting of samples with correct final answers
- **Suboptimal Data**, comprising samples with incorrect answers.

Each category contained 50,000 samples.

**Implementation and Comparison** We initialized both the policy and Q-function models using the Qwen-2.5-1.5B-instruct model. These models were then trained using Low-Rank Adaptation (LoRA) with the following hyperparameters: rank=8, lora_alpha=32, and dropout=0.1. We evaluated performance on the **unseen** GSM8k test set. This evaluation compared QTPO against direct behavior cloning, using models trained on various mixtures of expert and imperfect data.

**Results** The results of these comparisons are presented in the figure 4b. QTPO consistently outperformed behavior cloning across all settings, achieving an average accuracy improvement of **1.81%**. This consistent enhancement is particularly noteworthy as our method relies exclusively on a static offline dataset.

### D.4 COMPARISON WITH SOME BASELINES

To ensure a rigorous evaluation, we further benchmark QTPT against strong offline RL baselines, including Implicit Q-Learning (IQL) and Conservative Q-Learning (CQL), as well as the recent Transformer-based approach, Q-SFT. We evaluate these methods on the Linear Bandit task under both expert and random pretraining datasets.

As shown in Figure 6, QTPT significantly outperforms all baselines by a large margin. Specifically, on the expert dataset, QTPT achieves a cumulative regret of 39.4, whereas IQL, CQL, and Q-SFT

suffer from much higher regrets of 224.5, 161.2, and 208.0, respectively. This trend is even more pronounced on the random dataset, where QTPT maintains a low regret of 30.9, demonstrating its superior robustness to suboptimal data. In contrast, the baselines fail to learn effective policies, with regrets exceeding 200. These results highlight QTPT's ability to effectively leverage in-context learning for decision-making, surpassing both traditional conservative offline RL methods and supervised fine-tuning approaches.

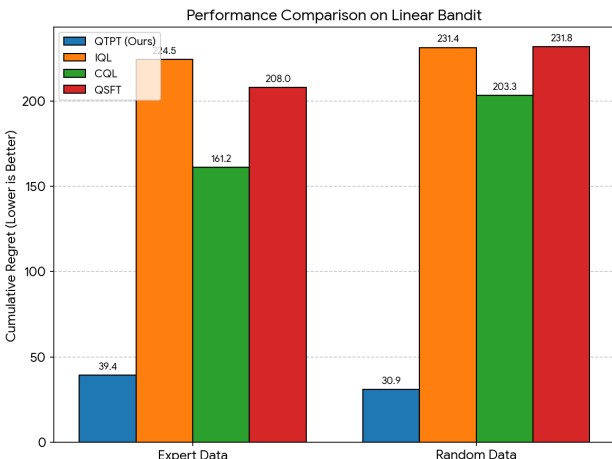

Figure 6: Performance comparison against strong baselines on the Stochastic Linear Bandit task

