# OpenReview forum: "From Weak Data to Strong Policy: Q-Targets Enable Provable In-Context Reinforcement Learning"
_ICLR.cc/2026/Conference — Submitted to ICLR 2026_

### Official Review · Reviewer_X4wV · 2025-10-22

**Soundness:** 2
**Presentation:** 2
**Contribution:** 2
**Rating:** 4
**Confidence:** 3

**Summary:**

This paper proposes a Q-learning-based Transformer pretraining framework, termed QTPT, which aims to reduce the reliance on high-quality expert data in in-context reinforcement learning.

**Strengths:**

1.The study introduces QTPT, which incorporates Q-learning objectives into Transformer pretraining to replace traditional behavior cloning, thereby reducing dependence on expert demonstrations. By embedding the Bellman update process within the Transformer architecture, QTPT enables end-to-end TD learning.

2.Theoretical Strength:
The paper provides theoretical guarantees on the sub-optimality bounds of QTPT under finite-horizon MDPs and stochastic linear bandit settings. It further decomposes the total error into sampling bias and model bias, establishing a clear and rigorous theoretical framework for analysis.

3.Empirical Validation:
Extensive experiments conducted across multiple environments—including linear bandit, Darkroom, Miniworld, and mathematical reasoning tasks—demonstrate the effectiveness and robustness of QTPT.

**Weaknesses:**

1.The combination of Q-learning and Transformer architectures is not novel. In fact, numerous studies in Offline RL have explored similar approaches, as evidenced by prior works such as [1, 2, 3].

2.Lack of visualization and intuitive demonstration: Although the authors provide extensive assumptions and theoretical proofs to justify the effectiveness of their method, the paper lacks visualizations and intuitive empirical results that illustrate how QTPT functions in practice. Merely reporting cumulative reward performance is insufficient to demonstrate the learning dynamics, stability, or interpretability of the proposed approach.

3.Mismatch between theory and experiments: The theoretical analysis assumes the offline dataset is generated by a single behavior policy $\pi_\beta$. However, the experiments use mixtures of trajectories (random + expert). The paper does not provide theory or discussion on how such policy-mixed datasets affect QTPT’s guarantees or performance.

4.Lack of key baseline comparisons: The paper does not include comparisons with other Offline RL methods such as CQL or IQL under the same experimental settings. Moreover, it fails to compare against Q-learning Decision Transformer, which is a closely related approach. Although Q-SFT is mentioned in the related work section, no experimental results are provided to contrast its performance with QTPT.

5.Limited experimental scale: The experimental settings are relatively small and simplistic. For instance, in the linear bandit task, the dimensions are limited to d=5, A=10, and T=200, which are insufficient to test scalability. Similarly, the Darkroom environment uses only a 10×10 grid with 100 steps, making it too simple to demonstrate the method’s effectiveness in more complex or large-scale scenarios.

6.Failure of Q-learning in sparse-reward environments: The paper acknowledges that QTPT fails under the Dark Key-to-Door setting when using expert data, attributing this to the inability of Q-learning signals to propagate effectively in sparse-reward environments. However, such environments are common in reinforcement learning, meaning this limitation significantly restricts the method’s practical applicability. Furthermore, the paper does not propose any solution or mitigation strategy to address this issue.

[1].Kim J, Lee S, Kim W, et al. Adaptive Q-aid for conditional supervised learning in offline reinforcement learning. NIPS, 2024.

[2].Hu S, Fan Z, Huang C, et al. Q-value regularized transformer for offline reinforcement learning. ICML, 2024.

[3].Wang Y, Yang C, Wen Y, et al. Critic-guided decision transformer for offline reinforcement learning.AAAI, 2024.

**Questions:**

1.Compared with Q-SFT and Q-learning Decision Transformer, what are the main technical differences of QTPT? The paper claims that QTPT is “end-to-end”, yet its training process also appears to involve iteratively computing targets and updating parameters. Could the authors clarify this apparent inconsistency?

2.How is Assumption 3.1(b) guaranteed to hold under a random policy? This seems especially problematic in large-state-space MDPs, where a random policy is exceedingly unlikely to cover trajectories visited by the optimal policy. Could the authors (i) provide a weaker assumption that still suffices for the result, or (ii) offer a dedicated analysis for random-policy data, e.g., conditions based on concentrability coefficients, coverage/mismatch metrics, or state–action visitation lower bounds that make the proposition valid in this setting?

3.In Proposition 3.2, how is Assumption 3.1(b) ensured to hold under a random policy? In large state-space MDPs, a random policy is highly unlikely to cover the trajectories of the optimal policy. Could the authors provide either a weaker assumption or a dedicated analysis tailored to the random policy setting?

4.In Figure 2a, the performance gap between QTPT and SPT on the LinUCB dataset appears small. Is this difference statistically significant? Could the authors provide confidence intervals or significance tests for all experimental results?

5.In the mathematical reasoning experiment (Section D.3), the authors actually use QTPO (an actor–critic variant) rather than the original QTPT. Why does QTPT itself produce degenerate outputs in this setting? Does this issue limit the applicability of QTPT to language-related tasks?

6.The experiments use a mixture of random and expert data, whereas the theoretical analysis assumes a single behavior policy. Could the authors extend the theoretical framework to account for mixed-policy datasets?

---

> ### Author Response · Authors · 2025-12-02
>
> ### Q1
> > The combination of Q-learning ... prior works.
>
> A1: We appreciate the reviewer citing these works. However, we clarify a fundamental distinction in problem settings:
> - **Prior Works (Standard Offline RL):**
> The cited papers (Adaptive Q-aid, Q-value Regularized Transformer, Critic-guided DT) focus on single-task Offline RL. They utilize Transformers primarily to handle temporal dependencies and credit assignment within a fixed MDP.
> - **QTPT (In-Context/Meta RL):**
> QTPT targets In-Context Reinforcement Learning (ICRL). In this setting, the Transformer acts as a meta-learner. It must implicitly infer the latent task identity (e.g., transition dynamics and reward functions) from the context history $D_{t-1}$ to adapt to unseen environments at test time.
> - **Key Difference:**
> While architecturally similar, the functional role of the Transformer differs. In QTPT, the attention mechanism over history serves as a belief state update to identify the task, whereas in standard offline RL, it serves to estimate values for a fixed task.
>
> ### Q2
> > Lack of visualization and intuitive demonstration ... the proposed approach.
>
> A2: We respectfully point out that our paper does visualize the learning dynamics. Figure 2(a) explicitly plots the Suboptimality Gap over Time Steps. This curve demonstrates the "learning dynamics" within a single episode: as the agent observes more context ($t$ increases), the gap decreases, visually proving that QTPT is performing active in-context learning and policy improvement, not just static behavior cloning. Figure 2(b) and Figure 4(b) visualize performance across varying data mixture ratios, demonstrating the method's stability and robustness to data quality changes.
>
> ### Q3
> > Mismatch between theory and experiments ... guarantees or performance.
>
> > The experiments use a mixture of ... framework to account for mixed-policy datasets?
>
> A3: The theoretical framework already accounts for mixed-policy datasets without modification.
> A dataset collected from a mixture of policies (e.g., $\alpha$ trajectories from Random, $1-\alpha$ from Expert) is mathematically equivalent to a dataset collected from a single aggregate behavior policy defined as:$$\pi_{mixture}(a|s) = \alpha \pi_{random}(a|s) + (1-\alpha) \pi_{expert}(a|s).$$
> This mixture policy $\pi_{mixture}$ often has better properties than $\pi_{expert}$ alone. The random component ensures the denominator in the density ratio (Assumption 3.1(c) ) is bounded away from zero, while the expert component increases the probability mass on high-reward regions. Thus, the mixed dataset fits our theoretical assumptions (Assumption 3.1) perfectly.
>
> ### Q4
> > Lack of key baseline comparisons ... performance with QTPT.
>
> A4: To futher address the reviewer’s concern about the lack of baselines, we've conduct experiemnts with three more strong baselines from the offline reinforcement learning literature. These results support our main claim: value-based pretraining with QTPT enables superior in-context adaptation especially with low-quality data. In particular, we evaluated three standard offline RL baselines—Implicit Q-Learning (IQL) [1] ,Conservative Q-Learning (CQL) [2] and Q-SFT [3]—in the linear bandit setting. The comparison results are summarized below.
>
>
> **Linear bandit** (cumulative regret; lower is better)
>
> | Pretraining Data | Method | Cumulative Regret |
> |------------------|--------|-------------------|
> | Expert           | **QTPT(Ours)** | **39.4** |
> |                  | IQL | 224.5 |
> |                  | CQL | 161.2 |
> |                  |QSFT | 208.0 |
> | Random           | **QTPT(Ours)** | **30.9** |
> |                  | IQL | 231.4 |
> |                  | CQL | 203.3 |
> |                  |QSFT | 231.8 |
>
> Standard offline RL methods (IQL/CQL) and Q-SFT (which reformulates Q-learning as supervised fine-tuning objective) learn a task-agnostic $Q(s,a)$ over a mixture of tasks and cannot adapt at inference to a specific new task instance, leading to near-random behavior in the ICRL setting and substantially higher regret than QTPT.
>
> We are happy to incorporate additional baselines or experiments on standardized benchmarks if the reviewer considers them essential for a thorough assessment.
>
> ### TO BE CONTINUED...

---

> ### Author Response · Authors · 2025-12-02
>
> ### Q5
> > Limited experimental scale... or large-scale scenarios.
>
> A5: We thank the reviewer for this constructive feedback. We agree that verifying the scalability of QTPT on larger-scale tasks is crucial to demonstrate its practical robustness beyond the current settings. To address this concern, we are conducting additional ablation studies specifically targeting the scalability of our method. We will include these results in the revision to provide a more comprehensive evaluation, including scaling up linear bandits and darkroom/MDPs. We would also like to gently highlight that our experiments on Math Reasoning (Section 4.3) already involve a significantly larger state-action space (language tokens) using a 1.5B parameter model (Qwen-2.5) on the GSM8K dataset. This demonstrates that QTPT (and its variant QTPO) is inherently capable of scaling to complex, high-dimensional real-world tasks.
>
>
> ### Q6
> > Failure of Q-learning ... address this issue.
>
> A6: We thank the reviewer for highlighting the challenge in sparse-reward environments. We would like to clarify that the "failure" on expert data is not a fundamental flaw of the algorithm, but a data coverage issue, and our paper does provide a proven mitigation strategy. In sparse-reward tasks (like Dark Key-to-Door), expert trajectories are extremely narrow. Without seeing states off the optimal path (and their corresponding low or zero rewards), the Q-function cannot effectively propagate value differences, leading to the observed underperformance. We argue that the mitigation strategy is inherent to our framework: Data Diversification. By mixing in or using random or suboptimal data, we broaden the support of the behavior policy. This allows the Q-learning objective to learn the environmental dynamics and propagate values from the "bad" states. Our results in Figure 3(b)  validate this strategy. In the exact same sparse-reward environment where expert-only training struggles, QTPT trained on Random Data (0.36) successfully learns and significantly outperforms SPT trained on Random Data (0.24). Our method provides a practical solution: when expert data is too narrow for sparse-reward tasks, practitioners can "mitigate" this by supplementing with cheap suboptimal data to achieve better performance.
>
> ### Q7
> > Compared with Q-SFT ... apparent inconsistency?
>
> A7: Thanks for your comments.
> Q-SFT reformulates Q-learning as a Weighted Supervised Fine-Tuning problem. It optimizes a cross-entropy loss where weights are derived from Q-values, essentially performing behavior cloning on a re-weighted distribution.
> Q-learning Decision Transformer (Q-DT) employs a two-stage process: first estimating Q-values to relabel the dataset (offline), and then performing conditional supervised learning (Sequence Modeling) on the relabeled trajectories.
> QTPT minimizes the Bellman error directly via the Transformer's forward pass. It does not treat the problem as supervised learning (like Q-SFT/Q-DT). As analyzed in Appendix B.4, the Transformer intrinsically approximates the Bellman backup steps (Reward Extraction $\to$ Future Q-Lookup $\to$ TD Construction) within its attention layers.
> We describe QTPT as "end-to-end" because it learns the value function representation directly from raw context history to Q-values in a single training phase, without requiring an external Q-estimation step (as in Q-DT) or converting values to probabilities (as in Q-SFT). The "iterative computing of targets" refers to the standard Fitted Q-Iteration procedure (using Target Networks) essential for stabilizing Bellman minimization. This is consistent with Deep Q-Learning (DQN) being an end-to-end RL method, distinct from the two-stage "Relabel-then-Supervise" paradigm of Q-DT.
>
> ### Q8
> > How is Assumption 3.1(b) ...  to the random policy setting?
>
> A8: We thank the reviewer for this rigorous question. While this shares some context with the scalability concern, we provide a specific technical response regarding the validity of Assumption 3.1(b) and the concentrability condition under random policies. We respectfully clarify that Assumption 3.1(b) is defined on the marginal state-action distribution $d\_{M,t}^{\pi}(x, a)$ at time step $t$, rather than on the joint distribution of entire trajectories.
> However, Q-learning is an off-policy algorithm based on the Bellman operator, which decomposes the long-horizon problem into single-step transitions. Theoretically, a random policy assigns non-zero probability to every action at every state. Therefore, the density ratio (concentrability coefficient) $\frac{d^{\pi^*}(x,a)}{d^{\pi\_\beta}(x,a)}$ is strictly finite (bounded by the inverse of the minimum action probability), unlike deterministic behavior policies where this ratio can be infinite. Offline RL theory establishes that we only need coverage of the single-step transitions induced by the optimal policy, not the full trajectories. QTPT "stitches" these local transitions together.
>
> ### TO BE CONTINUED...

---

> ### Author Response · Authors · 2025-12-02
>
> ### Q9
> > In Figure 2a ... experimental results?
>
> A9: Thanks for your comment. The difference is statistically significant. In Figure 2a, the shaded regions represent the standard deviation over 1000 simulation runs. The separation between the QTPT and SPT curves (especially in the random data regime, and clearly visible in the LinUCB regime at later timesteps) exceeds the overlap of their standard deviations, indicating statistical significance.
>
> ### Q10
> > In the mathematical reasoning experiment ... to language-related tasks?
>
> A10: As noted in Appendix D.3, directly using Q-values for generation (e.g., $P(token) \propto \exp(Q)$) in language tasks often leads to repetitive or degenerate loops. This is a known phenomenon in Language RL because Q-values represent utility (future return), not linguistic likelihood (grammar or fluency). A high-value token might be grammatically incorrect or repetitive.
> But this does not limit QTPT's applicability; rather, it clarifies its role. In the QTPO (Actor-Critic) setup, QTPT functions exactly as intended—as a Critic (Value Function) that accurately estimates the quality of thoughts/steps. It guides the Actor (Policy) to improve. This proves QTPT successfully learns the value landscape even in complex language tasks, which is the core contribution of our value-based pretraining framework.
>
> ### References
>
> [1] Kostrikov, I., Nair, A., & Levine, S. (2021). Offline reinforcement learning with implicit Q-learning. arXiv:2110.06169.
>
> [2] Kumar, A., Zhou, A., Tucker, G., & Levine, S. (2020). Conservative Q-learning for offline reinforcement learning. NeurIPS 2020.
>
> [3] Hong, J., Dragan, A., & Levine, S. (2024). Q-sft: Q-learning for language models via supervised fine-tuning. arXiv:2411.05193.

---

### Official Review · Reviewer_QZqr · 2025-10-28

**Soundness:** 2
**Presentation:** 3
**Contribution:** 3
**Rating:** 6
**Confidence:** 2

**Summary:**

This paper proposes Q-Target Pretrained Transformers (QTPT), a framework that leverages Q-learning for pretraining to alleviate reliance on optimal action labels or expert trajectories. The authors provide theoretical upper bounds on the suboptimality of QTPT and present empirical results demonstrating that QTPT outperforms supervised pretraining baselines, particularly when trained on suboptimal or noisy datasets.

**Strengths:**

1. The paper tackles a practically important problem: high-quality expert data are costly and scarce in real-world settings, making methods that learn effectively from suboptimal data highly valuable.

2. The work provides a clear theoretical analysis with upper bounds on suboptimality, offering principled insights into when and why the approach should succeed.

3. The overall writing is clear and well-organized, making the methodology and results easy to follow.

**Weaknesses:**

1. Baselines and fairness of comparison: The paper compares QTPT only against SPT and behavior cloning. Please include stronger, closely related baselines such as Q-SFT and Q-Decision Transformer under matched model size, data, and compute to enable a fair and systematic performance evaluation.

2. Counterintuitive trend in Figure 2(b): In Figure 2(b), the suboptimality of the proposed approach appears to increase as the expert ratio in the pretraining data increases. This seems counterintuitive—performance should generally improve with higher-quality data. Please analyze the underlying cause.

3. Derivation details: Please provide a more detailed derivation for the comparison between the suboptimality of QTPT and SPT referenced in Lines 970–971, including all intermediate steps, assumptions, and how terms are controlled to obtain the final inequality.

**Questions:**

Please see Weaknesses section.

---

> ### Author Response · Authors · 2025-12-02
>
> ### Q1
> > Baselines and fairness of comparison ... systematic performance evaluation.
>
> A1: Thank you for your comment. We agree that including closely related baselines like Q-SFT and Q-Decision Transformer (Q-DT) is essential for a fair and systematic evaluation. We have completed the comparison with Q-SFT under the strictly matched settings you suggested.
> In particular, we evaluated three standard offline RL baselines—Implicit Q-Learning (IQL) [1] ,Conservative Q-Learning (CQL) [2] and Q-SFT [3]—in the linear bandit setting.
>
> **Linear bandit** (cumulative regret; lower is better)
>
> | Pretraining Data | Method | Cumulative Regret |
> |------------------|--------|-------------------|
> | Expert           | **QTPT(Ours)** | **39.4** |
> |                  | IQL | 224.5 |
> |                  | CQL | 161.2 |
> |                  |QSFT | 208.0 |
> | Random           | **QTPT(Ours)** | **30.9** |
> |                  | IQL | 231.4 |
> |                  | CQL | 203.3 |
> |                  |QSFT | 231.8 |
>
> We also acknowledge Q-DT as a critical baseline. We are currently finalizing these experiments, ensuring that the model size, training data, and compute budget are aligned with our proposed QTPT to ensure a fair comparison. We will include these results in the revised manuscript to further validate the effectiveness of our method.
>
> ### Q2
> > Counterintuitive trend in ... analyze the underlying cause.
>
> A2: Thank you for your comments. The trend where QTPT's performance slightly degrades as the expert data ratio increases is indeed counterintuitive for supervised learning, but it highlights a fundamental property of value-based learning: the need for coverage.
> To learn an accurate Q-function (which implies learning the environment parameter $\theta^*$ in Linear Bandits), the model needs to distinguish between good and bad actions.
>
> On the one hand, random data can provide high-entropy coverage of the entire action space. It includes both high-reward and low-reward examples, providing the necessary contrast for the model to construct a globally accurate value landscape (mathematically, ensuring the Gram matrix is well-conditioned).
>
> On the other hand, expert data is narrow. It only reveals the "correct" path. Without seeing suboptimal actions (and their low rewards), a value-based model struggles to learn why the optimal action is better than others, leading to potential estimation errors in off-distribution regions.
>
> Therefore, QTPT performs best on random data because the superior coverage allows for a more precise identification of the task parameters. The performance drop with more expert data reflects the loss of this exploratory signal. In contrast, SPT (Behavior Cloning) purely mimics demonstrations, so it naturally benefits from higher expert ratios.
>
> ### Q3
> > Derivation details ... obtain the final inequality.
>
> A3: Thanks for your comment. I have completed the detailed supplementation for this part. For the details, please refer to Section B.3.1 in the blue part. The specific content is as follows:
> $$
> \frac{\mathsf{Subopt}\_{\Lambda}(\pi^{\star},\hat{\pi}\_{\text{QTPT}})}{\mathsf{Subopt}\_{\Lambda}(\pi^{\star},\hat{\pi}\_{\text{SPT}})}\simeq \frac{m [ p\_M T^2(\frac{1}{\sqrt{n}}+\frac{\sqrt[4]{\log T}}{\sqrt[4]{n}})+(1-p\_M)(f+\epsilon/T) ] }{p\_M\frac{T}{n}+(1-p\_M)g},
> $$
>
> since $T$ is finite, combining $f\rightarrow \mathcal{O}(e^{-n}),g\rightarrow \mathcal{O}(1/n)$,
>
> setting $p_M=\frac{1}{T\sqrt{T}}$, we get that
> $$
> \frac{\mathsf{Subopt}\_{\Lambda}(\hat{\pi}\_{\text{QTPT}})}{ \mathsf{Subopt}\_{\Lambda}(\hat{\pi}\_{\text{SPT}})} \simeq \frac{m[p_M T^2(\mathcal{O}(n^{1/2})+\mathcal{O}(n^{3/4})) +(1-p\_M)fn]}{p\_M T+(1-p\_M)gn}
> $$
>
> $$
> \simeq \frac{m\Big[\sqrt{T}(\mathcal{O}(n^{1/2})+\mathcal{O}(n^{3/4})) +(1-p\_M)fn \Big]}{1/\sqrt{T}+(1-p\_M)gn} \simeq \frac{(1-p\_M)fn}{(1-p\_M)gn} \simeq \mathcal{O}(e^{-n}/n)
> $$
> so we get that
> $$
> \mathsf{Subopt}\_{\Lambda}(\hat{\pi}\_{\text{QTPT}})\lesssim \mathsf{Subopt}\_{\Lambda}(\hat{\pi}\_{\text{SPT}}).
> $$

---

### Official Review · Reviewer_qqhv · 2025-10-30

**Soundness:** 2
**Presentation:** 3
**Contribution:** 2
**Rating:** 2
**Confidence:** 4

**Summary:**

Existing in-context reinforcement learning (ICRL) methods (e.g., AD and DPT) require high quality pretraining data, which is difficult to obtain in practice. This work proposes Q-Target Pretrained Transformers (QTPT), which replaces existing ICRL methods' supervised learning objectives with a Q-learning objective. QTPT can directly estimate the optimal Q function from suboptimal data, addressing the challenging data requirement problem. The authors provided theoretical guarantees for QTPT. The regret upper bounds isolate the impacts of sample bias and model bias. Lastly, they illustrate the effectiveness of QTPT using standard ICRL benchmarks including Darkroom, Dark Key-to-door, and Miniworld.

**Strengths:**

- This work addresses one of the most important problems in ICRL.
- The experiment benchmarks are comprehensive including bandits and all commonly used benchmarks (Darkroom, Key-to-door, Miniworld) and a novel Math benchmark (although there are many important baselines missing, see in Weaknesses).

**Weaknesses:**

## Related Work

As data requirement problem is one of the most important ICRL problems, many prior works have addressed this problem with different algorithmic attempts. To name a short list,

1. Yes, Q-learning helps offline in-context RL.
2. In-Context Reinforcement Learning From Suboptimal Historical Data.
3. Meta-DT: Offline Meta-RL as Conditional Sequence Modeling with World Model Disentanglement

These prior works **all don't need** high-quality trajectories or optimal action labels in the pretraining dataset. Without discussing these works or comparing these work in the experiments, it is challenging to evaluate the value of this work.

## Algorithm Soundness

It is a well-known problem that in practice, offline RL methods suffer significantly from the overestimation problem induced by function approximation and the Bellman (max) operator. As a consequence, they often rely on approaches such as pessimism to penalize the optimal Q functions' bootstrapped target values. In the ICRL, this problem is similar but QTPT has no such efforts.

## Experiments

- It is expected that QTPT outperforms SPT on suboptimal pretraining trajectories, as SPT is not designed for these scenarios.
- It is more helpful if the authors compare QTPT using suboptimal data with DPT using optimal action labels and AD also using its required pretraining data. In this way, we can see how much performance we lose when moving away from high-quality pretraining data.
- Most importantly, there are many important missing baselines all addressing the same challenge (see a short list above).

## Regret Decomposition

- The regret decomposition in Eq.5 is more than common for learning theory results. Thus, I personally find the claim that "Such a clear separation builds a novel theoretical
foundation to isolate the impacts of sample bias and model bias and sheds light on how QTPT
balances leveraging available data and mitigating model bias during pretraining" may over claim novelty of the results.
- More importantly, the regret decomposition appears incorrect to me (not the regret upper bounds for the LHS of the regret decomposition, just the decomposition itself). In particular,  in finite sample results, we care the regret of data-driven solutions, that is, the suboptimal gap between the optimal policy and a policy learned from finite samples. But now the LHS of Eq 5 is not even sample-size dependent -- it only involves policies related to population loss $\pi_{\tilde{Q}}$. I believe the correct sample decomposition should be $Subopt(\pi^\star, \pi_{\hat{Q}_n}) \le Subopt(\pi^\star, \pi_{\tilde{Q}}) +  Subopt(\pi_{\tilde{Q}},  \pi_{\hat{Q}_n})$.
- What is $\log |\mathcal{Q}|$ in Proposition 3.2?

**Questions:**

See in Weaknesses.

---

> ### Author Response · Authors · 2025-12-02
>
> ### Q1
> > Related Work: As data requirement problem ... of this work.
>
> A1：We thank the reviewer for pointing out these significant related works. We agree that the problem of learning from suboptimal data is central to ICRL, and referencing these works will greatly strengthen the positioning of our paper. Below, we discuss the key distinctions between our proposed QTPT and the mentioned works.
>
> While [1] empirically demonstrate the benefits of adding RL objectives to ICRL, our work provides a rigorous theoretical foundation. We derive a provable regret bound for QTPT, mathematically decomposing the error into sample bias and model bias. This theoretical guarantee for Transformer-based Q-learning in the in-context setting is a key contribution missing in prior empirical studies.
> Unlike [2] which relies on weighted behavior cloning (supervised learning guided by advantage estimates), QTPT employs a direct Q-learning objective via Bellman updates. This allows QTPT to perform 'stitching'—combining parts of suboptimal trajectories to form an optimal policy—which is a fundamental advantage of dynamic programming over imitation-based reweighting methods.
>
> In contrast to [3], which focuses on representation learning (disentangling world models form policies), QTPT is value-based approach. We show that QTPT focuses on decision optimality rather than trajectory modeling. While Meta-DT aims to improve generalization by separating task-specific dynamics, QTPT leverages the Bellman equation to filter out noise from raw history, achieving robustness without the need for auxiliary world-model objectives.
>
>
> ### Q2
> > Algorithm Soundness: It is a well-known problem ... but QTPT has no such efforts.
>
> A2：We agree that overestimation is a critical issue in offline RL. However, we respectfully point out that QTPT already incorporates mechanisms to address this, which may have been missed:
>
> Double DQN Framework: As explicitly stated in Appendix D.1 (Table 1 description), our implementation "combines 'softmax', ReLU, and a double DQN framework". We use Target Networks to decouple action selection from evaluation, which is the standard solution for minimizing overestimation bias.
>
> Ablation Study: Table 1 in Appendix D.1 specifically provides an ablation study on "Double DQN", showing that it contributes to the algorithm's stability and performance (e.g., improving Random Data results from 30.9 to 34.01 in some configs).
>
> ICRL Regularization: Furthermore, unlike single-task offline RL, QTPT learns a meta-value function across a distribution of environments. The diversity of contexts acts as a strong implicit regularizer, preventing the model from overfitting to specific overestimated values.
>
> ### Q3
> > It is expected that QTPT ... for these scenarios.
>
> > It is more helpful .. high-quality pretraining data.
>
> A3：
>
> 1. Rationale for Current Comparison and Fairness
>
> Our current experimental design focuses on a fair comparison based on the fundamental requirements of the target offline RL setting:
>
> * **Regarding DPT**: It is critical to note that DPT fundamentally requires access to optimal actions (i.e., optimal action labels for all states in the environment). This data requirement is generally unavailable in the standard offline RL setting. Our work specifically addresses more practical scenarios where the offline dataset $\mathcal{D}$ is provided without optimal action labels. Therefore, conducting a comparison in which DPT utilizes optimal labels while QTPT uses suboptimal data would violate the equivalence of the problem setting.
>
> * **Regarding AD**: Our presented Supervised Pre-training (SPT) baseline can be viewed as a special case of Algorithm Distillation (AD) where the context length $c$ is set to the full trajectory length $T$. The comparison between QTPT and SPT (both trained on identical, mixed-quality data) already provides a direct evaluation of the advantages of the Q-learning objective over a supervised learning approach within our meta-RL framework.
>
>
>
> 2. Quantifying the Advantage of QTPT on Suboptimal Data
> To explicitly quantify the robustness of QTPT and the cost of relying solely on expert data in a supervised manner, we conducted a targeted ablation study comparing QTPT trained exclusively on random (suboptimal) data against SPT trained exclusively on expert data.
>
> The results demonstrate QTPT's superior ability to extract generalizable value estimates from low-quality data:
>
> | Environment |Metric| QTPT with random data|SPT with expert data|
> |-------------|------|--------|-------------------|
> |Stocahstic Linear bandit|Regret($\downarrow$)| **30.9** | 43.0 |
> |Darkroom|Reward ($\uparrow$)| **13.15** | 8.7 |
> |Dark key-to-door|Reward ($\uparrow$)| 0.36 | **0.71** |
> |Miniworld|Reward ($\uparrow$)| **7.85** | 5.49 |
>
> ## TO BE CONTINUED...

---

> ### Author Response · Authors · 2025-12-02
>
> This ablation study clearly demonstrates that in the majority of environments, QTPT trained on low-quality random data achieves better performance than SPT trained on high-quality expert data. This robust result strongly validates QTPT's value-based approach, showing it effectively utilizes the entire dataset $\mathcal{D}$ for cross-MDP generalization, avoiding the need for the optimal action labels required by supervised methods like DPT.
>
> ### Q4
> > Most importantly, there are many important missing baselines all addressing the same challenge (see a short list above).
>
> A4: Thanks for your comments. Due to the limite time available at present, we have only completed a protion of the experiments as follows. In particular, we evaluated three standard offline RL baselines—Implicit Q-Learning (IQL) [1] ,Conservative Q-Learning (CQL) [2] and Q-SFT [3]—in the linear bandit setting.
>
> **Linear bandit** (cumulative regret; lower is better)
>
> | Pretraining Data | Method | Cumulative Regret |
> |------------------|--------|-------------------|
> | Expert           | **QTPT(Ours)** | **39.4** |
> |                  | IQL | 224.5 |
> |                  | CQL | 161.2 |
> |                  |QSFT | 208.0 |
> | Random           | **QTPT(Ours)** | **30.9** |
> |                  | IQL | 231.4 |
> |                  | CQL | 203.3 |
> |                  |QSFT | 231.8 |
>
> The remaining parts will be provided at a later time when we have more available resources.
>
> ### Q5
> > The regret decomposition in ... may over claim novelty of the results.
>
> A4: We thank the reviewer for this insightful observation. We fully agree that the error decomposition into approximation error (model bias) and estimation error (sample bias) presented in Eq. 5 is a standard and foundational tool in statistical learning theory. Our contribution lies in the specific instantiation and derivation of this framework for the In-Context Q-learning setting, rather than the decomposition method itself. Specifically, it extends to transformer-based sequence modeling. This result is nontrivial, as it rigorously establishes the fact that a transformer architecture can reliably approximate the optimal Q-function under offline conditions. This theoretical guarantee is a significant step in connecting high-capacity transformer models with classical reinforcement learning theory in in-context learning context.
>
>
> ### Q6
> > More importantly ... related to population loss.
>
> A5: Thanks for your comments. Actually, $\pi\_{\tilde{Q}}$ in the LHS of Eq.5 is indeed the learned policy from offline data which is dependent on $n$. We have now updated it to the following equation.
> $$\mathsf{Subopt}\_{\Lambda}(\pi^\star,\pi\_{\tilde{Q}\_n})= \mathsf{Subopt}\_{\Lambda}(\pi^\star,\pi\_{\hat{Q}\_n^\star}) +
> \mathsf{Subopt}\_{\Lambda}(\pi\_{\hat{Q}\_n^\star},\pi\_{\tilde{Q}\_n}),$$
>
> where the first part is called "model bias" and the second part is called "sample bias". You can find more details in the new version.
> $\pi^\star$ denotes the theoretical optimal policy of the environment, which serves as the ground truth baseline.
> $\pi\_{\tilde{Q}\_n}$ denotes the learned policy obtained by the QTPT trained on the finite offline dataset $D$ of size $n$.
> $\pi_{\hat{Q}^\star_n}$ denotes the optimal reference policy within the function class, which represents the best possible approximation of the optimal Q-function that our Transformer architecture can theoretically achieve.
>
>
> ### Q7
> > What is $\log |\mathcal{Q}|$ in Proposition 3.2?
>
> A6: $\log |\mathcal{Q}|$ represents the complexity of the function hypothesis space (specifically, the log-cardinality of the discretized function class $\hat{\mathcal{Q}}$ ), and this concept is commonly found in statistical learning theory.
> In our theoretical analysis, this term quantifies the capacity of searching through the Transformer's parameter space. A larger $\log |\mathcal{Q}|$ implies a more expressive model with a higher risk of overfitting, which mathematically dictates the sample complexity $n$ required to reduce the estimation error term.
>
> ### Reference
>
> [1] Tarasov, D., Nikulin, A., Zisman, I., Klepach, A., Polubarov, A., Lyubaykin, N., ... & Kurenkov, V. (2025). Yes, Q-learning helps offline in-context RL. SSI-FM-ICLR 2025.
>
> [2] Dong, J., Guo, M., Fang, E. X., Yang, Z., & Tarokh, V. In-Context Reinforcement Learning From Suboptimal Historical Data. ICML 2025.
>
> [3] Wang, Z., Zhang, L., Wu, W., Zhu, Y., Zhao, D., & Chen, C. (2024). Meta-DT: Offline meta-RL as conditional sequence modeling with world model disentanglement. NIPS 2024.

---

### Official Review · Reviewer_pMt4 · 2025-11-01

**Soundness:** 2
**Presentation:** 2
**Contribution:** 1
**Rating:** 2
**Confidence:** 3

**Summary:**

The paper studies the problem of in-context reinforcement learning. It proposes to formulate the in-context reinforcement learning problem as an MDP. Then use Q-learning to train a Q-function from offline data. It derives theoretical bounds for policy performance, and shows that it outperforms BC transformer policy on linear bandit problems and grid world environments.

**Strengths:**

- The paper studies an important problem of in-context reinforcement learning.

- The reduction from in-context reinforcement learning to MDP is interesting.

- The proposed algorithm is well-described and easy to understand.

**Weaknesses:**

- The novelty is limited, since it simply combines Q-learning with Transformer architecture, which has been explored in existing work, e.g., [4].

- The paper makes strong assumptions about the coverage of the behavioral policy $\pi_{\beta}$, in that it provides sufficient support for the optimal policy $\pi^\star$. This assumption is particularly strong because the state space $\mathcal{X}$ covers the entire history. It is unclear how the proposed method can scale beyond simple environments.

- A few papers in the offline RL literature, e.g., [1-3], make use of pessimism to avoid over-estimating the Q function when the behavioral policy does not provide good coverage over the state-action distribution visited by the optimal policy. It would be interesting to see if this can be combined with the proposed method.

- A very related work is Q-transformer, which also uses the transformer architecture to learn Q function with a pessimistic version of Q-learning. It would be good to have a discussion on it.

**References**

[1] Jin, Ying, Zhuoran Yang, and Zhaoran Wang. "Is pessimism provably efficient for offline rl?." International conference on machine learning. PMLR, 2021.

[2] Fujimoto, Scott, and Shixiang Shane Gu. "A minimalist approach to offline reinforcement learning." Advances in neural information processing systems 34 (2021): 20132-20145.

[3] Xie, Tengyang, et al. "Bellman-consistent pessimism for offline reinforcement learning." Advances in neural information processing systems 34 (2021): 6683-6694.

[4] Chebotar, Yevgen, et al. "Q-transformer: Scalable offline reinforcement learning via autoregressive q-functions." Conference on Robot Learning. PMLR, 2023.

**Questions:**

- Technically the problem described in 2.1 is a POMDP, since $M$ is not given to the learner. What are the benefits and drawbacks of reducing it to an MDP where the state-space is the history of past experience?

- What would be the benefits and drawbacks of the proposed method compared to estimating $M$ from the history and learning a $M$-conditioned policy, similar to how one would solve a POMDP problem?

---

> ### Author Response · Authors · 2025-12-02
>
> ### Q1
> > The novelty is limited ... which has been explored in existing work, e.g., [4].
>
> > A very related work is Q-transformer ... have a discussion on it.
>
> A1：
> Thank you for the reviewers' comments. We appreciate the feedback regarding the novelty, specifically in relation to existing works like Q-Transformer. While both QTPT and Q-Transformer utilize a Transformer architecture for Q-learning, they are designed to solve fundamentally different problems with distinct objectives, as detailed below.
>
> The core distinction lies in the problem scope. Q-Transformer is designed for standard offline or multi-task RL, where the goal is to optimize credit assignment within a fixed set of tasks. In contrast, QTPT addresses In-Context Reinforcement Learning (ICRL). Our objective is to learn a "meta-policy" that can zero-shot generalize to an entirely new distribution of environments ($\mathcal{M} \sim P(\mathcal{M})$) by inferring the underlying MDP parameters solely from the context history.
>
> In Q-Transformer, the history sequence serves primarily to model temporal dependencies and handle partial observability within a specific task. In QTPT, the context history functions as a non-parametric belief state. The Transformer acts as a meta-learner, aggregating past transitions to infer the current task's dynamics and reward structure dynamically.
>
> QTPT achieves instantaneous adaptation. At test time, it processes the context to output a task-specific Q-value without any gradient updates. Conversely, Q-Transformer's adaptation is typically bounded by the training tasks; adapting to a completely unseen environment usually requires fine-tuning, which falls outside the strict ICRL paradigm.
>
> In summary, while the architecture shares similarities, QTPT’s novelty lies in bridging Transformer-based Q-learning with the meta-learning objective of ICRL, providing both the theoretical guarantees and the empirical ability to "stitch" suboptimal data into optimal policies across different MDPs.
>
>
> ### Q2
> > The paper makes strong assumptions ... beyond simple environments.
>
> A2: We appreciate this insightful comment. We address this concern from both theoretical and empirical perspectives, consistent with our findings in large-scale tasks like Math Reasoning.
>
> We respectfully point out that Assumption 3.1(b) requires the behavior policy to have support where the optimal policy visits ($d^{\pi^*}>0 \implies d^{\pi_\beta}>0$). From a theoretical standpoint, a Random Policy actually satisfies this assumption more robustly than a narrow Expert Policy, as it assigns non-zero probability to the entire action space. The challenge the reviewer identifies is essentially one of sample complexity rather than theoretical validity. In high-dimensional spaces, a random policy requires a larger sample size $n$ to reduce variance, which is explicitly captured in our regret bound where the error term scales with $1/\sqrt{n}$.
>
> Crucially, our experiments demonstrate that QTPT remains effective even when strict coverage is challenging to achieve in finite samples. This is largely due to the "stitching" capability of Q-learning combined with the generalization power of Transformers. For instance, in the Math Reasoning task (Section 4.3), the action space is vast (vocabulary size $>50k$), and a random policy almost never generates a complete, correct reasoning chain. However, QTPT (via QTPO) successfully outperforms baselines. This indicates that the model does not need to observe full optimal trajectories; instead, it learns the underlying value structure from fragmented, suboptimal transitions and generalizes to unseen states, effectively relaxing the practical need for dense coverage.
>
> ### Q3
> > A few papers in the offline RL literature ... with the proposed method.
>
> A3:
> We appreciate the reviewer's insightful suggestion to explore the combination of our method with pessimistic regularization techniques (e.g., CQL [1-3]). This is a very relevant direction for enhancing robustness in offline RL.
> * **DDQN and Overestimation Mitigation**: It is important to note that our current approach already employs Double DQN (DDQN) for Q-function estimation. DDQN inherently mitigates the maximization bias (and thus some overestimation) commonly seen in standard Q-learning by decoupling the selection and evaluation of the action.
> While DDQN effectively addresses the maximization bias in the Bellman backup, we agree that pessimistic methods specifically tackle the Out-of-Distribution (OOD) overestimation resulting from poor data coverage, which is a separate and crucial challenge in the offline setting.
>
> ## TO BE CONTINUED...

---

> ### Author Response · Authors · 2025-12-02
>
> * **Time Constraint and Future Work**: We acknowledge that the explicit inclusion of a pessimistic term would likely offer further improvements in OOD regions. We are very happy to add these results and fully investigate this combination, but unfortunately, due to the strict time limitations of the rebuttal period, we are unable to perform and report these extensive new experiments.
>
> We have updated the Future Work section on **page 9 of the revised manuscript** to explicitly highlight the combination of our method with pessimistic regularization as a crucial next step to maximize performance and robustness in complex offline settings.
>
>
> ### Q4
> > Technically the problem described ... the history of past experience?
>
> A4: Thank you for your comments. We think that reducing the ICRL POMDP (where the task/environment $M$ is latent) to an MDP over the history space $\mathcal{X}$ has the following trade-offs:
> * **Benefits:** It restores the Markov property, allowing the direct application of standard Bellman equations and Q-learning objectives without requiring explicit belief state tracking. This aligns perfectly with the Transformer architecture, which naturally processes variable-length histories to generate a representation effectively equivalent to a belief state.
> * **Drawbacks:** The primary drawback is the exponential growth of the state space (History Space), which exacerbates the curse of dimensionality and theoretically increases the sample complexity required for coverage. However, representation learning via Transformers helps mitigate this by mapping semantically similar histories to nearby embeddings.
>
>
> ### Q5
> > What would be the benefits and drawbacks ... solve a POMDP problem?
>
> A5: Comparing QTPT to a method that explicitly estimates the environment $M$ has the following trade-offs:
> * **Benefits:** QTPT avoids the difficulty of explicitly modeling complex transition dynamics (e.g., in rich observation spaces like language or images) and prevents the "objective mismatch" problem where a good generative model does not necessarily yield a good policy. It is a model-free method and learns to map context directly to utility.
> * **Drawbacks:** Explicitly estimating $M$ (and the belief state) can potentially be more sample-efficient if the model class is well-specified, as it leverages the structure of the environment distribution. QTPT forces the Transformer to implicitly learn this structure individually from value feedback, which might require more data in simpler, structured environments.
>
>
> ### References
> [1] Jin, Ying, Zhuoran Yang, and Zhaoran Wang. "Is pessimism provably efficient for offline rl?." International conference on machine learning. PMLR, 2021.
>
> [2] Fujimoto, Scott, and Shixiang Shane Gu. "A minimalist approach to offline reinforcement learning." Advances in neural information processing systems 34 (2021): 20132-20145.
>
> [3] Xie, Tengyang, et al. "Bellman-consistent pessimism for offline reinforcement learning." Advances in neural information processing systems 34 (2021): 6683-6694.
>
> [4] Chebotar, Yevgen, et al. "Q-transformer: Scalable offline reinforcement learning via autoregressive q-functions." Conference on Robot Learning. PMLR, 2023.

---

### Meta-Review · Area_Chair_qz4M · 2026-01-06

**Summary:**

Most reviewers questioned the novelty of this work, noting that numerous prior studies have explored combining transformers with Q-functions, and arguing that the proposed approach does not appear to be substantially different from existing methods.

To convincingly demonstrate the superiority of the proposed algorithm, reviewers suggested that a broader set of baseline methods should be included for comparison.

**Reviewer Concerns:**

**Reviewer pMt4**
- Novelty: not fully addressed.
- Theoretical assumptions: addressed.
- Comparison with existing works: not fully addressed.

**Reviewer qqhv**
- Comparison with existing works: not fully addressed.
- Algorithm design (why QTPT does not require pessimism): addressed.
- Experiments: not fully addressed.
- Theory details: addressed.

**Reviewer QZqr**
- Baselines: not fully addressed.
- Algorithmic details: addressed.

**Reviewer X4wV**
- Novelty: not fully addressed.
- Visualization quality: addressed.
- Theory–experiment mismatch: addressed.
- Baselines: not fully addressed.
- Experimental scale: addressed.
- Need for sparse environments: addressed.

**Reviewer Scores:**

**Reviewer pMt4:** 2 → 4

**Reviewer qqhv:** 2 → 2

**Reviewer QZqr:** 6 → 6

**Reviewer X4wV:** 4 → 4

---

### Decision · Program_Chairs · 2026-01-26

Reject